# An uncertain future for the climate and health impacts of anthropogenic aerosols in Africa

Joe Adabouk Amooli[1], Marianne T. Lund[2], Sourangsu Chowdhury[2], Gunnar Myhre[2],
Ane N. Johansen[2], Bjørn H. Samset[2], and Daniel M. Westervelt[1,3]

[1]Lamont-Doherty Earth Observatory, Columbia University, New York, NY, United States of America
[2]CICERO Center for International Climate Research, Oslo, Norway
[3]NASA Goddard Institute for Space Studies, New York, NY, United States of America

**Correspondence to:** Daniel M. Westervelt, danielmw@ldeo.columbia.edu

## Abstract

Limited data availability and distinct regional characteristics of sources lead to a wide range of future aerosol emissions projections for Africa. Here we quantify and explore the implications of this spread for climate and health impact assessments. Using the Evaluating the Climate and Air Quality Impacts of Short-Lived Pollutants (ECLIPSE), the Shared Socioeconomic Pathways (SSPs), and the United Nations Environment Programme (UNEP) emission projections, we find high scenario diversity and regional heterogeneity in projected air pollution emissions across Africa. Baseline emissions also vary in their sectoral split. Using 10 different emissions pathways as input to the Oslo chemical transport model version 3 (OsloCTM3), we find that regionally-averaged annual-mean population-weighted $PM_{2.5}$ concentrations exhibit divergent trends depending on scenario stringency, with Eastern Africa $PM_{2.5}$ concentrations increasing by up to 6 µg m$^{-3}$ (37 %, SD ± 2.7 µg m$^{-3}$) by 2050 under the UNEP Baseline, SSP370, and ECLIPSE Current Legislation scenarios. In almost all cases, excess deaths increase substantially, with increases of up to more than 2.5 times compared to the baseline. We also find a net positive aerosol-induced radiative forcing across Africa in all scenarios by 2050 except two high-sulfur emission UNEP scenarios, with values ranging from 0.03 W m$^{-2}$ in SSP119 to 0.27 W m$^{-2}$ in SSP585. The wide spread in projected emissions and differences in sectoral distributions across scenarios highlights the critical need for accurate activity data and harmonization efforts in preparation for upcoming assessments such as the 7th Assessment Report of the Intergovernmental Panel on Climate Change.

# 1. Introduction

Emissions of anthropogenic aerosols, their precursors, and reactive gases are projected to undergo substantial changes in the coming decades, having a major impact on climate and air quality (Lund et al., 2019; Westervelt et al., 2020). In the United States (U.S.) and Europe, sulfur dioxide ($SO_2$), black carbon (BC), and organic aerosol (OA) emissions have been declining in recent decades (Leibensperger et al., 2012; Westervelt et al., 2015). In contrast, emissions have significantly increased in Africa and other countries in the Global South in recent decades (Fontes et al., 2017; Samset et al., 2019; Wang et al., 2023). In major emitting regions, aerosol-induced cooling has offset up to 1°C of surface warming since the pre-industrial era (Samset et al., 2019), and the sensitivity of regional climates to local reductions in aerosol emissions is high (Samset et al., 2018; Westervelt et al., 2020). Aerosols impact surface temperature not only by directly scattering or absorbing incoming solar radiation but also indirectly by altering cloud properties, including their brightness, lifespan, and, in the case of absorbing aerosols, by reducing cloud cover through the semi-direct burnoff effect (Albrecht, 1989; Twomey and S., 1977). Changes in regional emissions of scattering and absorbing aerosols also significantly affect both local and distant precipitation patterns (Westervelt et al., 2017, 2018). Considering the ambitious goals of the Paris Agreement, it is becoming increasingly essential to understand the impact of aerosols and other short-lived climate forcers on total anthropogenic radiative forcing (Forster et al., 2024; IPCC, 2023; Lund et al., 2019).

Alongside their climate impacts, aerosols, particularly fine particulate matter ($PM_{2.5}$), are associated with various negative health effects. Prolonged exposure to ambient $PM_{2.5}$ has been linked to increased mortality from cardiovascular and cerebrovascular diseases, acute lower respiratory infections, lung cancer, and adverse birth outcomes (Burnett et al., 2018; Chowdhury et al., 2020, 2022; Cohen et al., 2017). Ischemic heart disease (IHD) and stroke have been identified as major contributors to the health burden linked to air pollution (Fang et al., 2025). The current human health effects of $PM_{2.5}$ are substantial with a recent estimate of 8.1 million deaths per year globally (GBD, 2024). The future impacts of air pollutants will be influenced by both changes in emissions and shifts in demographics (Lund et al., 2019; Wells et al., 2024).

The health impacts of air pollution become more severe with age because aging reduces physiological resilience and increases susceptibility to chronic, non-communicable diseases (Pope, 2007; Shumake et al., 2013). Older adults, particularly those aged 60 and above, are especially vulnerable to air pollution (Yin et al., 2021). This age group is growing rapidly, especially in low-income and middle-income countries (LMICs), where pollution levels are typically higher than in wealthier nations (UN, 2017). In addition to older adults, young children, particularly those under five, are also highly vulnerable (Murray et al., 2020). Africa is projected to account for at least 40 % of the global population of children under the age of five by 2050 (KC and Lutz, 2017). At present, about 30 percent of $PM_{2.5}$-related deaths in Africa occur among children under five, and this rate has been increasing over the past decade  (Murray et al., 2020).

Africa exhibits distinct regional characteristics in aerosol emission sources, resulting in a wide range of potential future pollutant emission pathways, with some scenarios predicting significant increases in pollutants in key regions (Abera et al., 2020; Shindell et al., 2022; Turnock et al., 2020; Wells et al., 2024). Ongoing and anticipated future changes in Africa, including rapid economic development, rising urbanization, continued dependence on traditional biomass fuels, agricultural practices, limited access to advanced pollution control technologies, infrastructure expansion, and population growth, are expected to increase exposure to air pollutants, although

the magnitude and direction of these changes remain highly uncertain across future scenarios (Abera et al., 2020; Bauer et al., 2019; Chowdhury et al., 2020; Wells et al., 2024). Despite this, research on air pollution-related health impacts in Africa, particularly outdoor air pollution, is limited (Abera et al., 2020; UNEP, 2022; Wells et al., 2024) due to the scarcity of observational data (Katoto et al., 2019).

Global emission inventories, including the Community Emissions Data System (CEDS) (Hoesly et al., 2018), the Evaluating the Climate and Air Quality Impacts of Short-Lived Pollutants (ECLIPSE) inventory from the Greenhouse Gas-Air Pollution Interactions and Synergies (GAINS) model (Amann et al., 2011), and the Emissions Database for Global Atmospheric Research (EDGAR) (Crippa et al., 2018; Janssens-Maenhout et al., 2019), have improved our understanding of historical emissions trends, while global emission scenarios such as the Shared Socioeconomic Pathways (SSPs) (Riahi et al., 2017), have improved our understanding of future plausible trends. However, Africa remains critically understudied due to limited data. Previous studies focusing on the region, such as Shindell et al. (2022) and Wells et al. (2024), have primarily relied on the SSP scenarios, limiting their scope in capturing a broader range of plausible future emission trajectories. This knowledge gap is particularly pressing given Africa's accelerating development, substantial spatial variation in economic growth and industrialization, and the wide range of emission sources across the region. Addressing this gap, our study focuses on African aerosol and precursor emissions, examining their climate and health impacts under a broader set of scenarios that extend beyond the SSP scenarios. This work provides critical insights into Africa's unique emissions landscape and lays the groundwork for future research and policy interventions.

We explore the range in projections of mid-century anthropogenic air pollution levels in Africa, and the associated region-specific health impacts, resulting from 10 different pathways using the Oslo chemical transport model version 3 (OsloCTM3). We present estimates of the projected future regional radiative forcing of anthropogenic aerosols across these scenarios. We also evaluate the model's performance against surface and satellite observations. We build upon and expand on previous studies by incorporating simulations of $PM_{2.5}$, its associated radiative forcing, and health impacts across a comprehensive range of scenarios, including SSPs, ECLIPSE, and the United Nations Environment Programme (UNEP) Integrated Assessment of Air Pollution and Climate Change in Africa pathways, to explore a broad spectrum of future possibilities under plausible conditions. To achieve this, we use a chemical transport model with a more detailed representation of the chemistry involved in the formation of $PM_{2.5}$, compared to Earth System Models which must sacrifice detailed online chemical mechanisms for computational efficiency.

In Section 2, we provide an overview of the study domain and the OsloCTM3 model. We describe the emissions, simulations, surface and satellite observations, and the methodologies for assessing health impacts and calculating radiative forcing. Section 3 discusses the results, highlighting the significant variability and regional heterogeneity in projected air pollution emissions across Africa. We examine the sectoral contributions to BC and $SO_2$ emissions under increasing scenarios for 2020 and 2050, and the differences between these years. The section also discusses sub-regional and continental annual mean $PM_{2.5}$ changes between 2015-2018 and 2050 across all scenarios. Furthermore, we discuss the excess number of deaths attributable to $PM_{2.5}$ for 2015-2018 and 2050 under each scenario. In Section 4, we discuss the implications of

our results, while Section 5 presents the conclusions and the key findings from the study and their broader implications.

## 2. Methods

## 2.1 Study Domain

In this study, we divided the African continent into five sub-regions: Western, Central, Eastern, Southern, and Northern, as illustrated in **Figure 1**. These divisions are commonly used in regional analyses for ease of interpretation and to reflect the continent's diverse environmental, health, economic, and demographic characteristics (Djeufack Dongmo et al., 2023; Fenta et al., 2020;
Safo-Adu et al., 2023). Each region differs in population density and industrialization, which in turn influence air pollution levels, and public health outcomes (Djeufack Dongmo et al., 2023; Safo-Adu et al., 2023). The regional boundaries were adapted from the World Health Organization (WHO) Administrative regions for Africa (WHO, 2024), with minor adjustments to include all continental countries and ensure complete geographic coverage of the African
continent. For example, several North and East African countries that are geographically part of Africa but classified in the WHO Eastern Mediterranean Region were reassigned to their respective continental sub-regions to better reflect geographic boundaries (see Supplement for details).

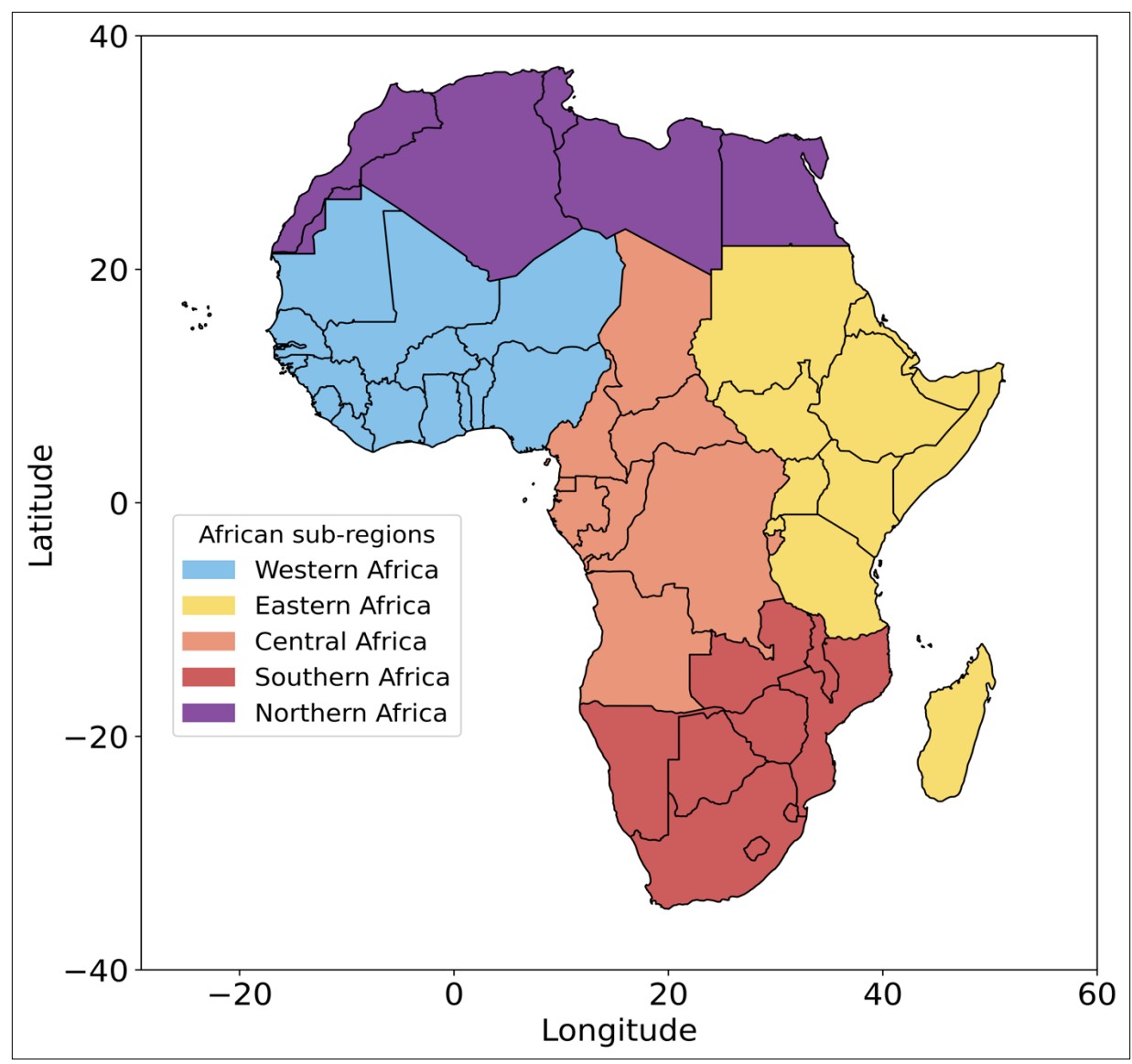


**Figure 1:** Study domain with regional definitions.

## 2.2 Emissions

We use both baseline (2015-2018) emissions and projected emissions for the year 2050 from ten available pathways:

- SSP1-1.9, SSP2-4.5, SSP3-7.0, and SSP5-8.5 (hereafter referred to as SSP119, SSP245, SSP370, and SSP585, respectively) (Riahi et al., 2017);
- ECLIPSE version 6b (ECL6) Current Legislation (ECL6 CLE), ECL6 Maximum
Feasible Reduction (ECL6 MFR), and ECL6 Sustainable Development Scenario (ECL6 SDS) (IIASA, 2024);

- UNEP Baseline (UNEP BASE), UNEP Short-Lived Climate Pollutants (UNEP SLCP), and UNEP Agenda 2063 (UNEP 2063) scenarios (UNEP, 2022).

Gidden et al. (2019) provides gridded SSP air pollution emissions from 2015 to 2100 across sectors, including residential, industrial, energy, transportation, waste, agriculture, solvents, and international shipping, at a spatial resolution of 0.5° latitude by 0.5° longitude. The SSP emissions include the following species: carbon dioxide ($CO_2$), methane ($CH_4$), carbon monoxide (CO), nitrogen oxides ($NO_x$), $SO_2$, non-methane volatile organic compounds (NMVOC), organic carbon (OC), and ammonia ($NH_3$). SSP119 represents a scenario with rigorous air

pollution control, low climate forcing, and minimal challenges in mitigation and adaptation, while SSP370 is marked by weak air pollution control, high climate forcing, and significant challenges in both mitigation and adaptation (Riahi et al., 2017). SSP245 serves as an intermediate pathway, while SSP585 is characterized by very high climate forcing and strong air pollution control, along with high socio-economic challenges for mitigation and low socio-economic challenges for

adaptation (O'Neill et al., 2017).

The ECLIPSE version 6b covers the years 1990 - 2050 and includes gridded aerosol and reactive gas emissions at 5 and 10-year intervals, provided at a spatial resolution of 0.5° latitude by 0.5° longitude. It also includes several other updates over earlier ECLIPSE versions, such as

improved regional resolution, particularly for Africa, updates to legislation and historical data, and revised gridding patterns for various sectors, including power plants, flaring, transportation, and industry (IIASA, 2024). ECL6 includes emissions for a wide range of species: $SO_2$, $NO_x$, $NH_3$, NMVOC, BC, OC, organic matter (OM), $PM_{2.5}$, $PM_{10}$, CO, and $CH_4$. The ECL6 CLE scenario projects future air pollution and climate change impacts based on the continuation of current

legislation and existing regulatory frameworks, while the ECL6 SDS scenario integrates ambitious air pollution controls with broader sustainable development goals, aiming to achieve significant co-benefits for health and climate. The ECL6 MFR scenario estimates the potential reductions in air pollution achievable through the implementation of all available advanced technologies and practices. We used 2016 ECL6 emissions as the baseline for calculating future changes in

ECLIPSE.

A recent UNEP report provides a comprehensive assessment of the interactions between air pollution and climate change in Africa (UNEP, 2022), bringing a unique focus on emissions scenarios specifically to Africa for the first time. This assessment utilizes the Low Emissions

Analysis Platform (LEAP) system, a prominent software tool for integrated energy policy planning, emissions reduction, and climate change mitigation evaluations (LEAP, 2021). The UNEP scenarios are provided at a spatial resolution of 0.5° latitude by 0.5° longitude, and include emissions of BC, $CH_4$, and hydrofluorocarbons (HFCs). We used the three UNEP scenarios: Baseline, SLCP, and Agenda 2063 to provide emissions for 2018 and 2050. The Baseline

scenario assumes that the energy, agricultural, and waste sectors will continue their current paths and that no new policies will be developed. In contrast, the SLCP scenario calls for the implementation of mitigation measures that specifically target warming SLCPs, such as BC (UNEP, 2022). Importantly, this scenario does not address $SO_2$ emissions, which have a cooling effect (UNEP, 2022). Most of the actions in this scenario are of a technological nature, such as

advances in technology and fuels. The Agenda 2063 scenario is a more ambitious scenario, which concentrates on mitigating measures that more widely accomplish the objectives of the Sustainable Development Goals (SDGs) and the African Union Commission (AUC) Agenda 2063 (UNEP, 2022). This scenario begins with the SLCP measures from the previous scenario and

then adds both technical and behavioral mitigating mechanisms that lead to greater
transformative change.

        These ten scenarios were selected to span a broad range of plausible futures, capturing variation in socioeconomic development trajectories (SSPs), explicit assumptions about air pollution policy ambition and implementation (ECLIPSE), and Africa-specific development priorities (UNEP). This
diverse set enables a comprehensive assessment of the potential climate and health impacts of anthropogenic aerosol emissions across Africa. All emissions datasets used share a common horizontal resolution of 0.5° latitude by 0.5° longitude, facilitating direct comparison and integration across scenarios. See Supplementary Table S1 for a summary of the emission scenarios and their characteristics.

        Since the UNEP emissions are provided only for the African domain, we used SSP245 emissions for the rest of the world. As part of this we also merged UNEP sector categories to align with those in the SSPs. Specifically, this means UNEP sectors residential, services, agricultural energy, and unspecified categories were merged into one residential category;
industry and industrial processes into industry; electricity, oil and gas, and charcoal into energy; and livestock and crop production into agriculture. Transport and waste are provided as individual categories in both SSP and UNEP scenarios. International shipping is not provided by UNEP, and we therefore used SSP245 emissions.

**2.3 OsloCTM3**
        OsloCTM3 is a global, three-dimensional Chemical Transport Model (CTM), driven by three-hourly meteorological forecast data from the European Centre for Medium-Range Weather Forecasts (ECMWF) Open Integrated Forecast System (OpenIFS) (Lund et al., 2018, 2019; Søvde et al., 2012). These forecasts are generated daily, starting with a 12-hour spin-up period
beginning from a noon analysis of the previous day, which are then combined to create a uniform dataset for an entire year (Søvde et al., 2012). The model is run in a 2.25° latitude by 2.25° longitude horizontal resolution and includes 60 vertical levels, with the highest level centered at 0.1 hPa. Although this horizontal resolution is relatively coarse, the OsloCTM3 model is well-suited for long-term, large-scale simulations of atmospheric composition. It includes detailed
aerosol chemistry and has been extensively applied in both global and regional studies (Aamaas et al., 2017; Lund et al., 2017, 2019; Søvde et al., 2012). OsloCTM3 simulates atmospheric concentrations of trace gases and all the main climate relevant aerosol species (black carbon, primary and secondary organic aerosol, secondary inorganic aerosol, sea salt and dust). The model description and evaluation of simulated aerosol distributions have been documented in
Lund et al. (2018, 2019).

        **2.4 Experiments**
        For each case, the model is run for 18 months, discarding the first 6 as spin-up. Meteorological data from the year 2010 OpenIFS were used for all simulations of the baseline
and future, with no feedback from climate change or variations in natural aerosols. This allows us to isolate the impact of anthropogenic emissions on air quality from the impacts of future climate changes on aerosols and trace gases. The model is initialized with fields including temperature, precipitation, relative humidity, pressure, cloud water, boundary-layer turbulence, advective and convective mass fluxes. We performed simulations using the ECLIPSE and UNEP
emissions. We use data from experiments with SSP119, SSP245, and SSP370 emissions

performed for Lund et al. (2019) and perform an additional simulation using SSP585 emissions for this study.

Results from simulations with future (2050) emissions are compared to simulations with the corresponding baseline emissions. The available emission scenarios have somewhat different baseline years: 2015 for the SSPs, 2016 for ECL6 emissions, and 2018 for UNEP emissions. The difference in simulated $PM_{2.5}$ concentrations between these three base years is 2 % on a continent-wide annual average (**Figure S3**), suggesting limited influence on our analysis from the differing time period and inventories considered. For this we use additional data from simulations with the CEDS21 emissions as input, to capture the most recent global emission inventory. These simulations were performed for and documented in Lund et al. (2023), however, neither model validation nor a dedicated Africa-focus was part of that study. $PM_{2.5}$ was calculated as the sum of individual fine-mode aerosol species, namely BC, primary and secondary organic aerosol (POA, SOA), sulfate ($SO_4$), dust, sea salt, nitrate ($NO_3$), and ammonium ($NH_4$). The fine-mode is defined with an upper cutoff at 2.5 µm, which corresponds directly to $PM_{2.5}$.

We analyzed the differences in population-weighted $PM_{2.5}$ between 2050 emissions and their respective baselines depending on the scenario as described above. Annual global population data at a 0.5° latitude by 0.5° longitude resolution, which is an input in the Inter-Sectoral Impact Model Intercomparison Project (ISIMIP2b) simulations (Jones and O'Neill, 2016), was used in our analysis. Both the OsloCTM3 $PM_{2.5}$ data and the population data were regridded to a 1° latitude by 1° longitude resolution using bilinear interpolation prior to calculating population-weighted values. The downscaling provides sufficient spatial resolution to capture major spatial variations in both $PM_{2.5}$ concentrations and population distributions. Comparison of modelled and observed surface $PM_{2.5}$ shows that the downscaled 1° latitude by 1° longitude resolution performs slightly better than the native 2.25° latitude by 2.25° longitude model resolution, when averaged over Africa. However, city-level performance varies by location (**Figure S1**). The population-weighted values were computed on a country-by-country basis by summing the products of population and $PM_{2.5}$ concentration at the grid level within each country and then dividing by the total population of that country (Chowdhury et al., 2019; Southerland et al., 2022). Grid cells were assigned to countries based on whether their center point lies within national boundaries, using a shapefile of administrative boundaries from the Natural Earth dataset (Natural Earth, 2025). SSP2 population data was used for UNEP and ECLIPSE cases, as SSP2 aligns closely with UN population projections (UNEP, 2022), while the respective SSP population projections were matched for the other SSP scenarios. To assess sensitivity to the population dataset, we repeated the calculations using SSP2 population data for all scenarios. The resulting population-weighted $PM_{2.5}$ values differed by less than 0.1 µg m$^{-3}$ on average, with no change to the relative scenario rankings or spatial patterns. This indicates that the population-weighted $PM_{2.5}$ estimates are not sensitive to the choice of SSP population dataset.

Using offline radiative transfer calculations with a multi-stream model using the discrete ordinate method (Myhre et al., 2013; Stamnes et al., 1988), we derived the AOD and the instantaneous top-of-the-atmosphere radiative forcing due to aerosol–radiation interactions resulting from changes in anthropogenic emissions. We also include an estimate of the radiative forcing of aerosol–cloud interactions using a parameterization based on Quaas et al. (2006) to account for the change in cloud droplet concentration resulting from anthropogenic aerosols, which alter the cloud effective radius and thus the optical properties of the clouds. The method has been used in several previous studies (e.g., Lund et al., 2019, 2023; Myhre et al., 2013,

2017). Note that this method excludes any contributions from cloud lifetime changes, which are typically estimated to be smaller than cloud albedo effects but not negligible (Stjern et al., 2016).

## 2.5 Observations

We evaluated the OsloCTM3, driven by CEDS21 anthropogenic emissions, against $PM_{2.5}$ surface observations from various United States embassy locations in Africa for the year 2019, as a first-order sanity check. The observed data was obtained from the AirNow database (U.S. EPA, 2024). The United States embassy monitors are high-quality reference instruments that provide publicly available, quality-assured $PM_{2.5}$ data. Located in major African cities, they offer reasonable spatial coverage of the most populated and emission-relevant regions in Africa and are often used as a benchmark for calibrating local monitoring networks. Additionally, we compared the model's $PM_{2.5}$ estimates with high-resolution satellite-derived $PM_{2.5}$ estimates, which were obtained from combined AOD retrievals from National Aeronautics and Space Administration (NASA) MODIS, Multi-angle Imaging Spectroradiometer (MISR), and Sea-viewing Wide Field-of-view Sensor (SeaWiFS) instruments, integrated with the Goddard Earth Observing System Chemistry (GEOS-Chem) transport model. These data are available in the Satellite-derived $PM_{2.5}$ Archive (Van Donkelaar et al., 2021). The model's performance was assessed using the coefficient of determination ($R^2$), root mean squared error (RMSE) and mean absolute error (MAE).

We evaluated the modelled AOD from CEDS21 against data retrieved from the MODIS instrument aboard the Aqua satellite, specifically the MYD08_D3_V6.1 release, which was accessed via the NASA Giovanni interface for the year 2019. The MYD08_D3 product is a gridded daily global dataset from MODIS (Platnick, 2015). It provides daily averaged values for atmospheric parameters on a 1° latitude by 1° longitude resolution, including aerosol properties, total ozone, atmospheric water vapor, cloud characteristics, and atmospheric stability indices (Platnick, 2015). For our analysis, we used the combined Dark Target and Deep Blue AOD at 550 nm from this release.

## 2.6 Health impact assessment

In combination with the generated data on ambient $PM_{2.5}$ exposure, we applied the MR-BRT (meta-regression-Bayesian, regularized, trimmed) exposure-response function (Murray et al., 2020; Pozzer et al., 2023), which was also used in previous studies (Chowdhury et al., 2022, 2024) and the most recent iteration of the Global Burden of Disease study (GBD, 2021). To account for uncertainty, we drew multiple samples from the posterior distribution of the exposure-response function. We assumed a theoretical minimum risk exposure level (TMREL) of 2.4 µg m$^{-3}$, held constant across all locations and scenarios (Murray et al., 2020). Below this threshold, no further reduction in mortality is assumed. The MR-BRT function was developed using global cohort data, including studies from high-exposure settings, such as Li et al. (2018), Yang et al. (2018), Yin et al. (2017), and Yusuf et al. (2020), enhancing its relevance for regions with high $PM_{2.5}$ levels, such as Africa (Chowdhury et al., 2022). Additionally, we incorporated uncertainty in baseline mortality rates reported by the respective countries. Full methodological details can be found in Chowdhury et al. (2020) https://iopscience.iop.org/article/10.1088/1748-9326/ab8334, and are not repeated here.

Using the cause-specific exposure-response function, we estimated excess deaths from ischemic heart disease (IHD), stroke (both ischemic and hemorrhagic), chronic obstructive pulmonary disease (COPD), lung cancer (LC), and Type II diabetes (T2DM) among adults (aged

25 and above), as well as acute lower respiratory tract infections (ALRI) among children (under 5 years old). The excess death burden was calculated at a 25 km by 25 km spatial resolution by interpolating the modeled ambient concentration data, and the estimates were stratified by age and disease category, following the approach of previous studies (Chowdhury et al., 2020, 2024):

$$M_{c,a,d} = P_a \times BM_{a,d} \times \frac{RR_{c,a,d} - 1}{RR_{c,a,d}}$$
(Eq. 1)

$RR_{c,a,d}$ is the Relative Risk, where c,a,d denotes the concentration of $PM_{2.5}$, population age, and disease, respectively; $BM_{a,d}$ is the baseline mortality rate per 100,000 population; and $P_a$ is the exposed population in a grid by age. Each variable is a function of specific dimensions:
RR varies with concentration of $PM_{2.5}$, age group, and disease; BM varies with age group and disease; and population varies by age group and location.

Excess deaths were estimated separately for adults and children at each 5-year interval. $RR_{c,a,d}$ were derived using MR-BRT functions for all diseases. Age specific RRs for IHD and stroke
are obtained using MR-BRT. For LC, T2-DM, and COPD, uniform $RR_{c,d}$ were used across all age groups among adults. $BM_{a,d}$ was obtained from the GBD (GBD, 2024) for all countries in Africa for the years 2015, 2016 and 2018. For 2050, we use the projected baseline under SSP scenarios mortality rates generated in a previous study by combining information from GBD and International Futures (Yang et al., 2023). We considered BM to remain uniform within a country
at 25 km by 25 km resolution by age and disease. Age distributions at 5-year intervals (adults > 25 years), and <5 years for children were obtained from the SSP database (Riahi et al., 2017) which are then merged with the gridded population data at 25 km by 25 km horizontal resolution under the respective SSP scenarios to obtain the age-specific population ($P_a$) at each 25 km by 25 km grid. For the concentration data generated using SSP emissions, we applied the respective
SSP projections for baseline mortality and population. For other emission scenarios (UNEP and ECL6), we utilize the SSP2 projections.

### 3. Results
#### 3.1 Projected air pollution emission trends
There is a large spread and regional heterogeneity in projected air pollution emissions across Africa and its sub-regions, as shown in **Figure 2** for BC and $SO_2$. Similar variability is seen for OC and $NO_x$ in Supplementary **Figure S4**. This variability suggests that the differences in local regulations, technological advancements, and economic development trajectories can substantially influence emission trends. As such, regional policy differences may play a crucial
role in reducing uncertainties in emissions projections. While we focus on the year 2050, we here show the full UNEP and SSP timeseries to illustrate that the spread in emissions continues through the century. We note that there are other scenarios in the ECL6 and SSP databases, but the ones considered in the present study span the large range shown in **Figure 2**.

BC emissions in Africa are projected to increase under SSP370, SSP585, SSP245, UNEP
Baseline, and ECL6 CLE by 2050, and to decrease under SSP119, UNEP SLCP, UNEP 2063, ECL6 MFR, and ECL6 SDS scenarios. After mid-century, BC emissions in Africa are projected to decline under the SSP245 and SSP585 scenarios, driven by a reduction in residential fossil fuel use (Turnock et al., 2020). The largest increases in BC emissions are projected under the UNEP Baseline and SSP370 scenarios, with increases of 2,116 kt/year (109 %) and 1,143 kt/year (47

%), respectively. In contrast, the largest declines in BC emissions are projected under the ECL6 SDS and SSP119 scenarios, with reductions of 2,146 kt/year (93 %) and 1,992 kt/year (82 %), respectively (**Table S3**). OC and $NO_x$ exhibit similar trends (**Figure S4**); however, OC decreases under the SSP585 scenario.

For $SO_2$ emissions, the overall pattern in Africa shows projected increases under all scenarios by 2050 except for the ECLIPSE scenarios, with declines after mid-century under SSP119, SSP245, and SSP585. The largest increases in $SO_2$ emissions are projected under the UNEP baseline and UNEP SLCP scenarios, with increases of 10,550 kt/year (142 %) and 9,095 kt/year (121 %), respectively, while the largest decreases are projected under the ECL6 SDS and ECL6 MFR scenarios, with reductions of 9,561 kt/year (87 %) and 7,011 kt/year (64 %),
respectively, by 2050. $SO_2$ emissions increase dramatically under UNEP SLCP by 2050 as this is a scenario focusing on reducing short-lived climate pollutants with an overall climate warming effect, such as BC—without addressing $SO_2$ emissions (UNEP, 2022). The UNEP SLCP scenario envisions a transition where 90 % of wood and charcoal use switches to efficient stoves by 2063 in urban areas, with similar shifts in rural areas. It also assumes a gradual transition from gas and
liquefied petroleum gas (LPG) to efficient electricity starting in 2030 (UNEP, 2022). The UNEP Agenda 2063 scenario shows a decline in both BC and $SO_2$ emissions by 2050. This scenario assumes a 1.4 % annual decrease in energy intensity for household energy use (compared to constant levels in the baseline), a 93 % improvement in refrigerator efficiency by 2063, and a doubling of air conditioner efficiency (UNEP, 2022). Additionally, it assumes that 30 % of
passenger kilometers shift from cars to buses, and 25 % switch to cycling.

Certain regions are projected to maintain high $SO_2$ emissions even under some stringent scenarios. For instance, in Central Africa and Southern Africa $SO_2$ emissions are projected to increase by 303% and 47% by 2050 under the SSP119 scenario, driven by reliance on coal during
the transition period (Wells et al., 2024). Under the SSP585 scenario, $SO_2$ emissions are projected to increase by 429% and 402% in Western Africa and Eastern Africa by 2050, driven by energy and industrial emissions (Shindell et al., 2022). By 2050, BC and $SO_2$ emissions in Northern Africa are projected to increase substantially under SSP370. The increases in sub-regional BC, $SO_2$, OC, and $NO_x$ emissions are projected to decline after mid-century under most
SSP scenarios, with the exception of SSP370, where emissions increase in Western Africa but continue to decrease in Southern Africa. Under the UNEP baseline scenario, BC, OC, $NO_x$ emissions are projected to be largest in Western Africa and Eastern Africa, followed by Central Africa, Southern Africa, and Northern Africa, by 2050, while $SO_2$ emissions are projected to be largest in Southern Africa, followed by Eastern Africa, Northern Africa, Western Africa, and
Central Africa (**Table S3**).

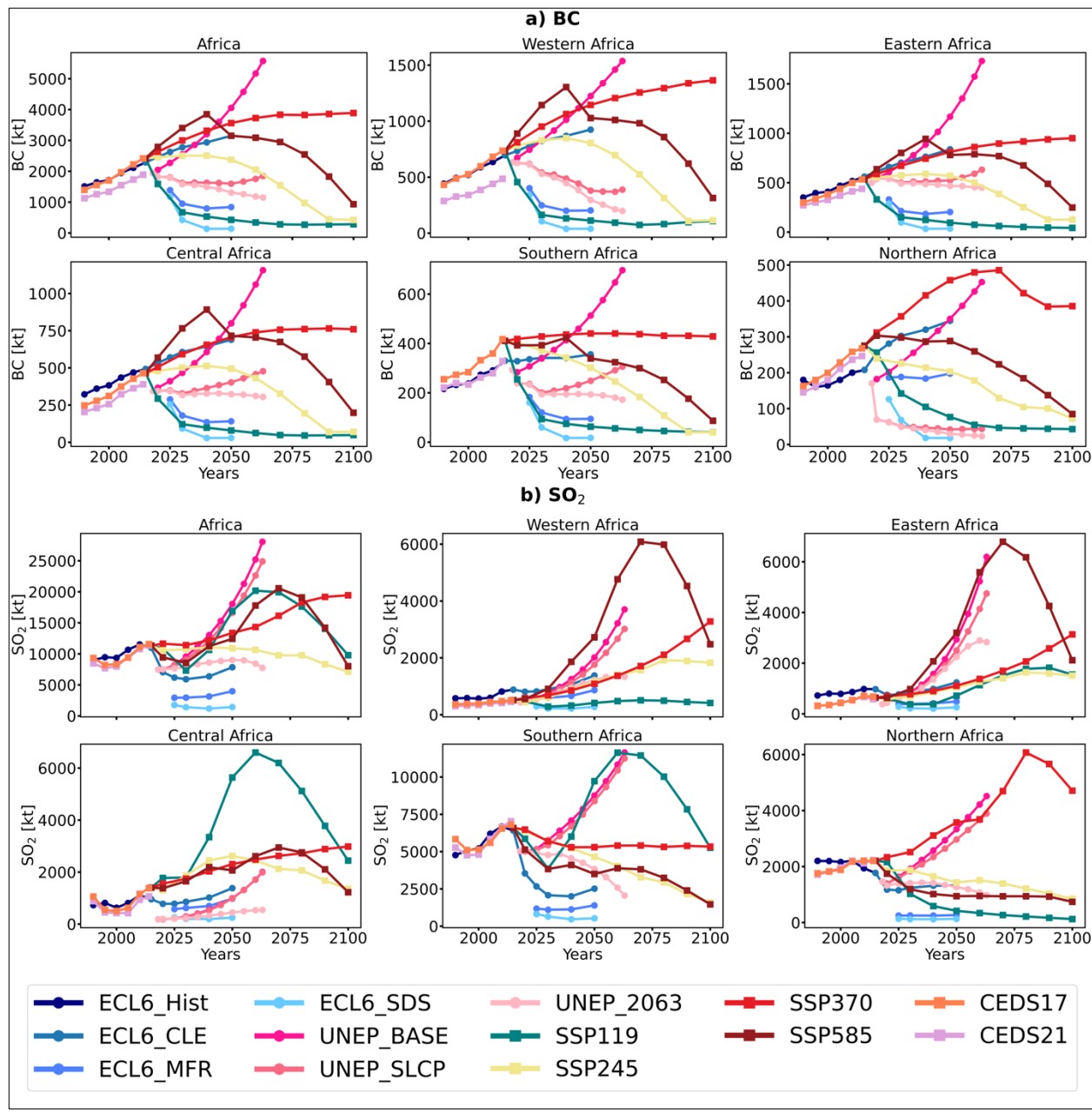

**Figure 2:** Large spread and regional heterogeneity in projected air pollution emissions in Africa. Here shown for a) BC and b) SO$_2$. Note: The y-axis scales vary between regions. ECL6_Hist represents historical emissions from the ECLIPSE version 6b inventory. CEDS17 and CEDS21 represent historical emissions from the CEDS versions 2017 and 2021, respectively.

**3.2 Sectoral contributions to air pollution emissions**

In many cases, the emissions source distributions are distinct, with notable discrepancies in sectoral shares across regions, as exemplified in **Figures 3 and 4** for the increasing scenarios SSP370, ECL6 CLE and UNEP Baseline emissions in 2020 and 2050. For instance, in Eastern Africa, the UNEP baseline scenario attributes 58 % of 2020 BC emissions to the residential sector, while the ECL6 CLE scenario attributes 75 %, and SSP370 attributes 84 %. Different socio-economic projections, technological advancements, and policies influence the regional emission sources and sectoral contributions. Consequently, the discrepancies in sectoral emission distributions among UNEP, SSPs, and ECL6 highlights the need for accurate activity data, emissions factors, and harmonization, particularly in the upcoming 7th Assessment Report of the Intergovernmental Panel on Climate Change (IPCC AR7). This is essential for ensuring consistency across integrated assessment models (IAMs) and for better constraining scenario baselines by grounding models in the most current and reliable emissions data and sectoral activity patterns.

The residential sector is the largest contributor to BC emissions across sub-Saharan Africa in all increasing scenarios in 2020 and 2050. In Northern Africa, BC emissions are dominated by the transportation sector across all increasing scenarios. The widespread use of clean cooking fuels, such as natural gas and Liquefied Petroleum Gas (LPG), by the majority of the population in Northern Africa (IEA, 2020) explains the comparatively lower residential emissions in this sub-region. Under SSP370, the residential sector will contribute 86 % in Western Africa, 84 % in Central Africa, 83 % in Eastern Africa, and 75 % in Southern Africa, by 2050. The second-largest contributor to BC emissions change unique to SSP370 is international shipping (5-12 %) which cannot be wholly attributed to African nations but may be linked to increasing trade associated with economic development. In Northern Africa, the transportation sector will contribute 61 % of the total BC increase under this scenario, followed by international shipping at 12 %.

By 2050, under the ECL6 CLE scenario, the transportation and waste sectors will be the second and third largest contributors to BC emissions in sub-Saharan Africa, following the residential sector. In Western Africa, 77 % and 8 % of BC emissions are projected to come from the residential sector and transportation sector by 2050, compared to 83 % and 5 % respectively, in 2020. The sectoral contributions to emissions changes (increases or decreases) in UNEP Baseline stand in sharp contrast with SSP370 which are mainly dominated by changes in household biomass cooking. For instance, in Western Africa, the transportation and residential sectors will account for 52 % and 27 % of the increase, respectively, in contrast with SSP370 which were mostly dominated by residential emission changes alone. **Figures S5** and **S6**, presented in the Supplement, illustrate sectoral contributions to BC emissions in 2020 and 2050, respectively, across the other scenarios. In sub-Saharan Africa, the residential sector remains the dominant source of BC emissions, while in Northern Africa, the transport sector contributes the most. Notably, under SSP119, SSP585, and the UNEP scenarios, sub-Saharan Africa shows an increasing share of BC emissions from the energy sector, indicating a growing influence of power generation as electrification expands across the region.

Similarly, the major sectoral contributors to $SO_2$ emissions in each sub-region of Africa are the industrial, energy, and transportation sectors, depending on the scenario (**Figure 4**). By 2050, under SSP370, industrial sector $SO_2$ emissions contribute 49 % in Western Africa, 86 % in Central Africa, 49 % in Eastern Africa, and 44 % in Northern Africa to total $SO_2$ emissions, which hints at a rapid northward shift in Africa's industrial center. To that end, Southern Africa's $SO_2$ emissions are projected to decrease under SSP370, with the industrial sector contributing 64 % and the

energy sector contributing 27 % to this reduction.   A similar shift in industrial and energy sector SO$_2$ emissions occurs in ECL6.  **Figures S7** and **S8** show sectoral contributions to SO$_2$ emissions in 2020 and 2050, respectively, across the other scenarios. The distribution of emissions is similar to that observed in the increasing scenarios, with Africa undergoing industrialization, except in Southern Africa.

Unlike SSP370, the UNEP Baseline scenario suggests a more distributed emissions profile across African sub-regions in 2050, similar to ECL6, particularly with regard to transportation emissions, which are more muted in SSP370. Under the UNEP Baseline scenario, the largest sectoral contributors to emissions are projected to be the transport, energy, and industrial sectors in Western Africa, Central Africa, Eastern Africa, and Northern Africa, while in Southern Africa, the energy and residential sectors are projected to be the largest contributors. Additionally, reductions in agricultural waste burning on fields in Africa by 2050 are projected under the UNEP Baseline scenario, as well as SSP370 and ECL6 CLE, due to the adoption of alternative agricultural practices, improved waste management, and emissions-reducing technologies.

In sub-Saharan Africa, across the increasing scenarios, OC emissions are predominantly from the residential sector, while NO$_x$ emissions are primarily driven by the transportation sector under SSP370 and the UNEP baseline scenarios, and by the residential sector under ECL6 CLE (see Supplement, **Figure S9** and **S10**).

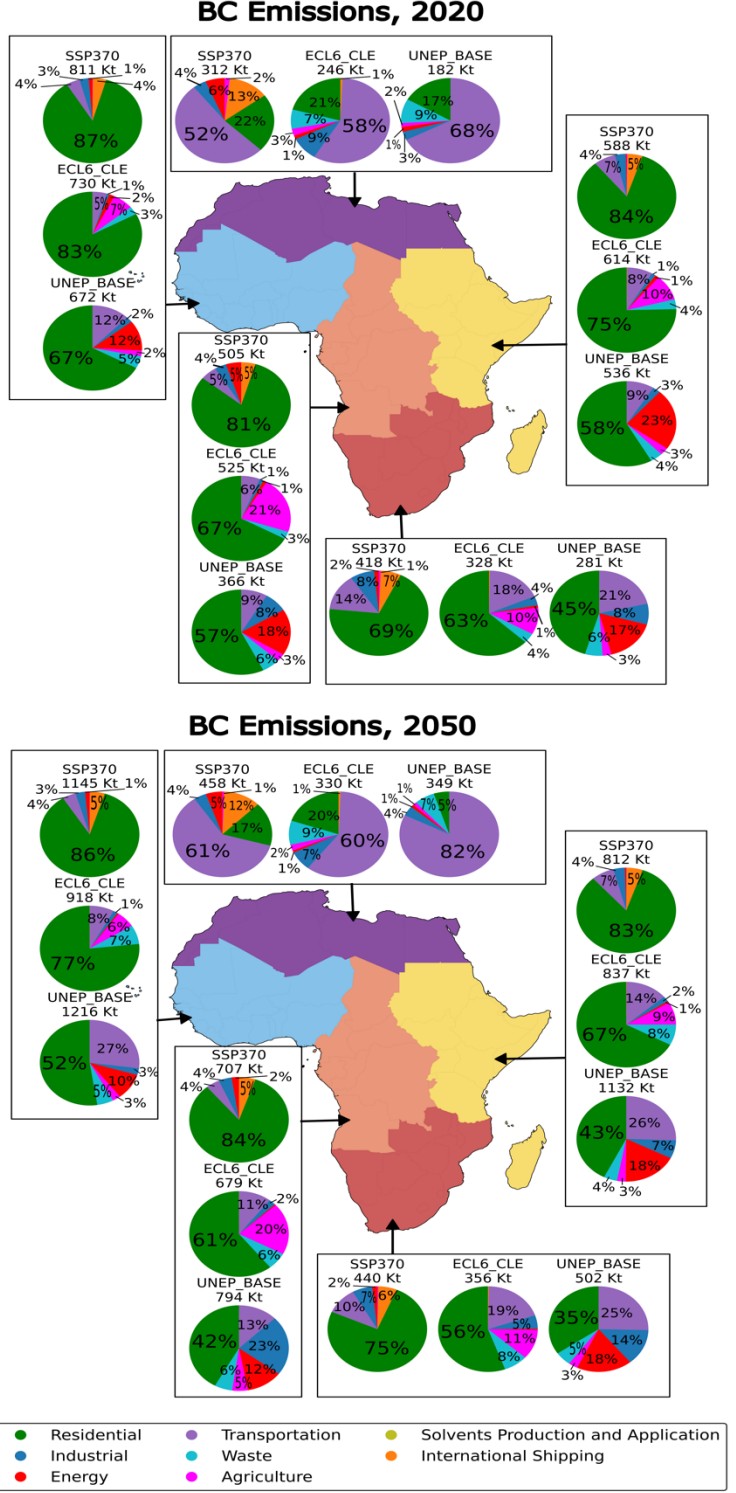

**Figure 3:** Sub-regional sectoral contributions to BC emissions (kilotons per year) in 2020 and 2050 under the increasing scenarios SSP370, ECL6 CLE, and UNEP Baseline.

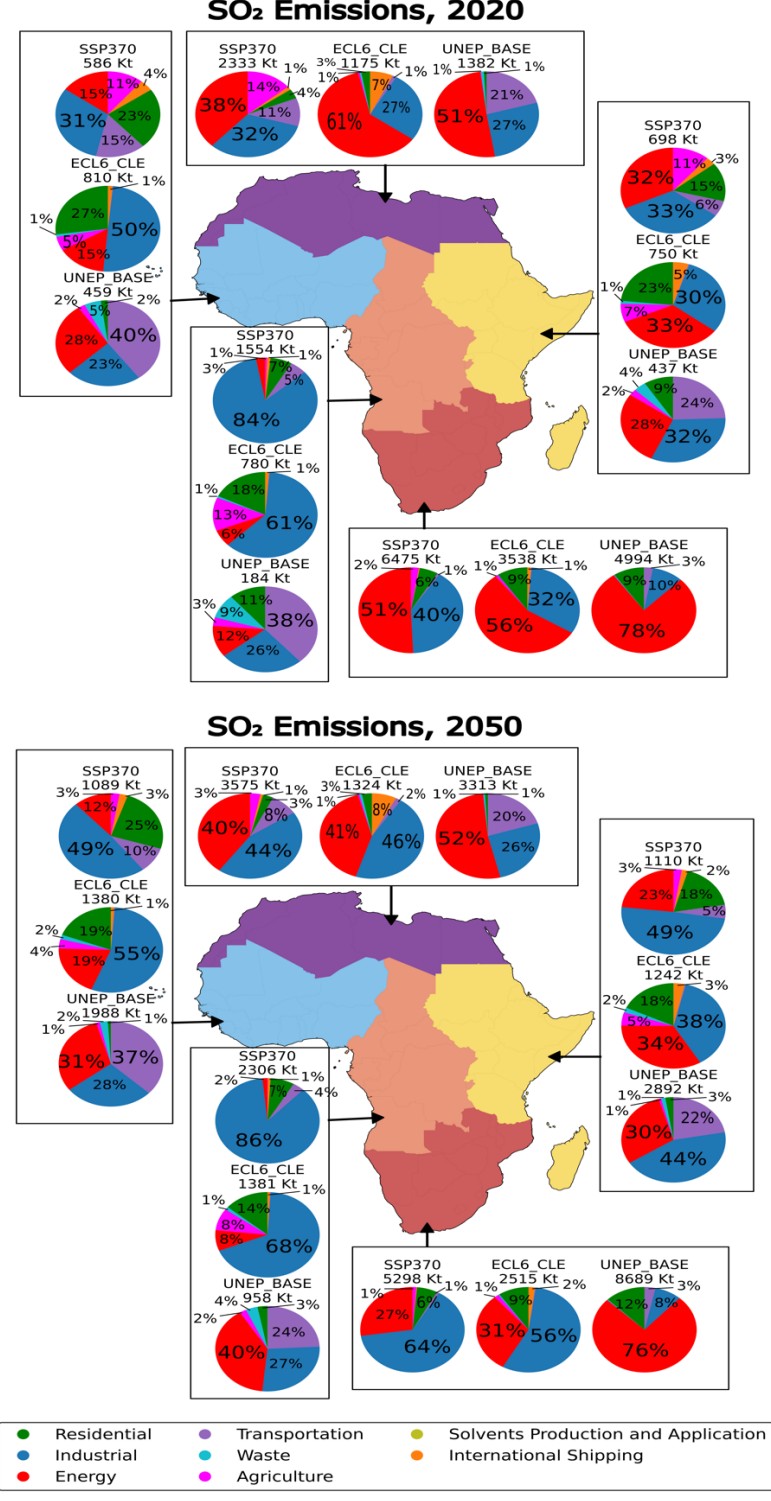

**Figure 4:** Sub-regional sectoral contributions to $SO_2$ emissions (kilotons per year) in 2020 and 2050 under the increasing scenarios SSP370, ECL6 CLE, and UNEP Baseline.

### 3.3 Annual-mean population-weighted PM$_{2.5}$ concentrations

We validated the OsloCTM3 model against PM$_{2.5}$ surface observations for 2019 and MODIS Aqua AOD data across Africa (see Sect. 2.5), finding a broad range of performance across regions. There is some disagreement between the observational datasets themselves, which should be considered when interpreting model performance. For PM$_{2.5}$ surface observations, R$^2$ values range from 0.48 to 0.76, and MAE values range from 5.7 µg m$^{-3}$ to 23.8 µg m$^{-3}$ in Western and Central Africa. For MODIS Aqua AOD, the validation yields an R$^2$ of 0.36, with RMSE and MAE of 0.11 and 0.08, respectively (see **Figure S1, Figure S2,** and **Table S2** for details). Overall, these validation results indicate that OsloCTM3 captures key spatial and temporal variations in PM$_{2.5}$ and AOD across Africa, though with varying performance across regions.

Given the model's reasonable performance for present-day, we now examine projected changes in future PM$_{2.5}$ levels. **Figure 5** shows annual-mean population-weighted PM$_{2.5}$ concentrations across Africa in 2050 under the highest emission scenario (UNEP Baseline) and the lowest emission scenario (SSP119), as well as their difference. Surface PM$_{2.5}$ concentrations differ by up to a factor of 2 between the highest and lowest scenario when averaged across the African continent in 2050, with markedly higher local spread. Notably, the most substantial differences between the UNEP Baseline and SSP119 are observed in Nigeria, Benin, Niger, Uganda, Ethiopia, Kenya, Democratic Republic of the Congo, and Egypt, where PM$_{2.5}$ concentrations differ by as much as 20 µg m$^{-3}$ (75 %) in 2050 projected populations. Most of these regions, which are characterized by high population density and industrial activity, experience elevated pollution in the UNEP scenario, highlighting the need for aggressive emission reduction measures.

**Figure 6** presents regionally-averaged annual-mean population-weighted (P.W.)  PM$_{2.5}$ concentrations in 2050 relative to the baseline. PM$_{2.5}$ concentrations are projected to increase under UNEP Baseline, SSP370, and ECL6 CLE scenarios, and decrease under the other scenarios, by 2050 when averaged across the African continent. The largest increase is projected under the UNEP Baseline scenario, with an increase by up to 6 µg m$^{-3}$ (37 %) in Eastern Africa, while the largest decrease is projected under the ECL6 SDS scenario, with a reduction by up to 5 µg m$^{-3}$ (8 %) in Western Africa and 3 µg m$^{-3}$ (39 %) in Southern Africa. Projected annual PM$_{2.5}$ concentrations in Africa follow a similar trend, with an increase of 4 µg m$^{-3}$ (15 %) under the UNEP Baseline scenario and a decrease of 5 µg m$^{-3}$ (20 %) under the ECL6 SDS scenario. Western, Eastern, and Northern Africa are projected to experience the most substantial PM$_{2.5}$ exposure increase under the UNEP Baseline, SSP370, and ECL6 CLE scenarios due to increased emissions from residential, industrial, transportation, and energy sectors. Conversely, the most substantial decreases are projected under the ECL6 SDS, ECL6 MFR, and SSP119 scenarios (**Figure 6 and Table S3**).

In Southern Africa, annual PM$_{2.5}$ levels are projected to decrease under the SSP370 and ECL6 CLE scenarios by 0.6 µg m$^{-3}$ (6 %) and 0.3 µg m$^{-3}$ (3 %), respectively, primarily due to reductions in energy and industrial emissions. However, Western and Eastern Africa show slight increases under the SSP245 scenario (1 %) and UNEP SLCP scenario (7 %), respectively. The most substantial increases in PM$_{2.5}$ exposure are projected in Nigeria, Egypt, Uganda, Rwanda, Burundi, and Benin under the UNEP Baseline and SSP370 scenarios, with concentration

increases ranging from 10 μg m$^{-3}$ to 20 μg m$^{-3}$ (11 % to 33 %), as shown in **Figure S12**.
Conversely, the largest decreases are observed in Nigeria under the ECL6 SDS, SSP119, and
UNEP 2063 scenarios, followed by significant reductions in Uganda, Rwanda, Burundi, and
Egypt. These observed differences largely reflect variations in PM$_{2.5}$ concentrations as well as
shifts in population distribution.

       In some areas the relative changes in PM$_{2.5}$ concentrations are small despite notable
absolute changes (**Figure 6**), which is attributed to naturally high baseline concentrations from
sources like dust, which cause air quality limits to already be exceeded (Pai et al., 2022). The
varying outcomes across scenarios emphasize the critical role of policy choices in determining
future air quality in Africa. The benefits in Nigeria, Uganda, Rwanda, Burundi, and Benin will
primarily result from improvements in the residential sector, which plays a dominant role in
reducing BC emissions. In countries such as South Africa, the benefits will largely stem from
reductions in industrial and energy sector emissions of SO$_2$. In Northern Africa, the benefits will
be driven primarily by reductions in emissions from the transportation sector.

       Generally, the results are consistent with the emissions changes (**Figs. 2-4**), in that
stringent energy and air quality policy scenarios such as ECL6 SDS, SSP119, ECL6 MFR, and
UNEP 2063 result in PM$_{2.5}$ decreases across the board, and more extractive and fossil-fuel
dominated scenarios (UNEP BASE, SSP370, and ECL6 CLE) result in worse future PM$_{2.5}$ air
quality. These results highlight the potential for effective policy interventions to mitigate air
pollution and improve public health across the continent.

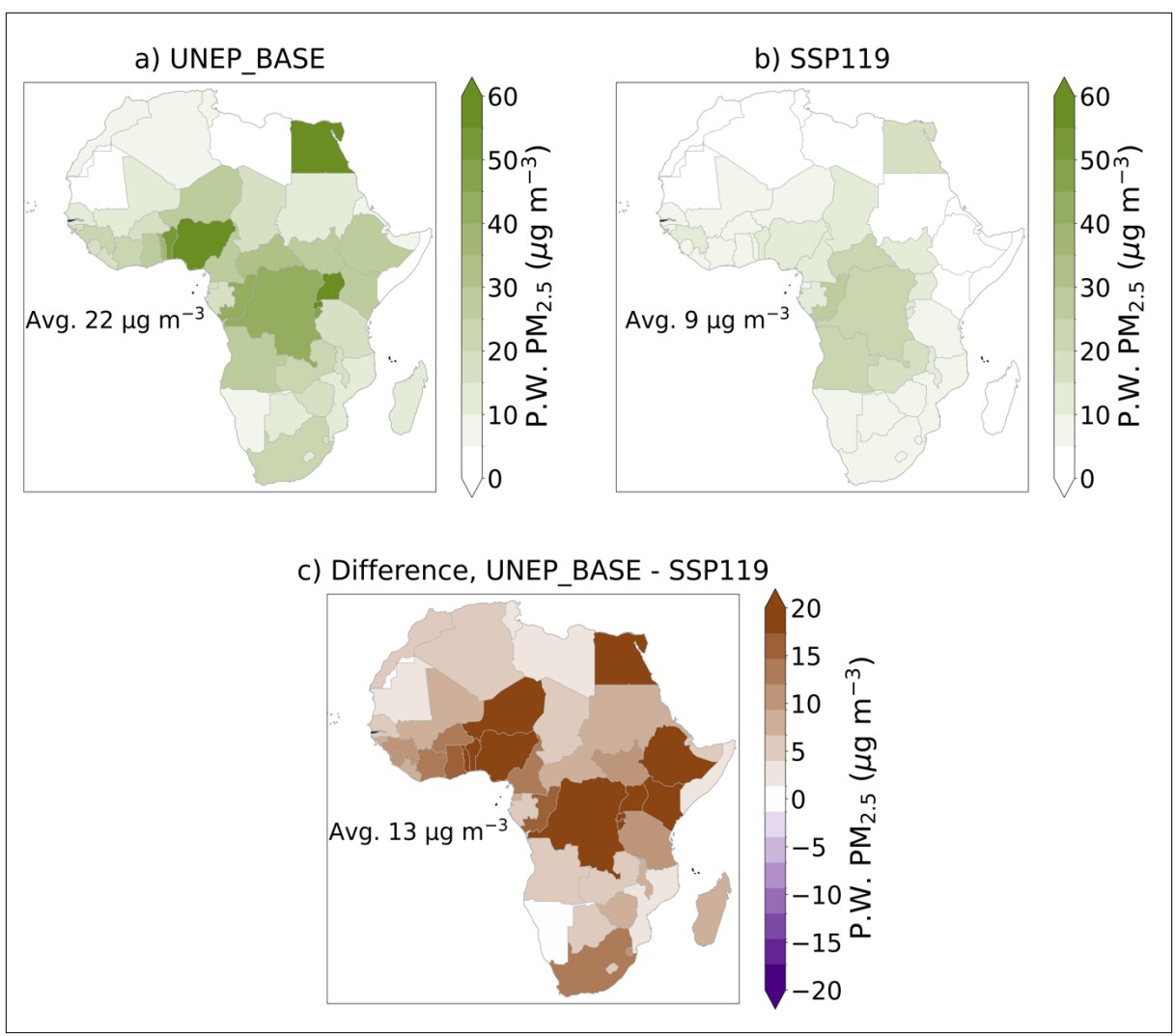

**Figure 5:** Annual-mean population-weighted PM$_{2.5}$ concentrations across Africa in 2050 under a) the highest emission scenario (UNEP Baseline), b) the lowest emission scenario (SSP119), and c) their difference.

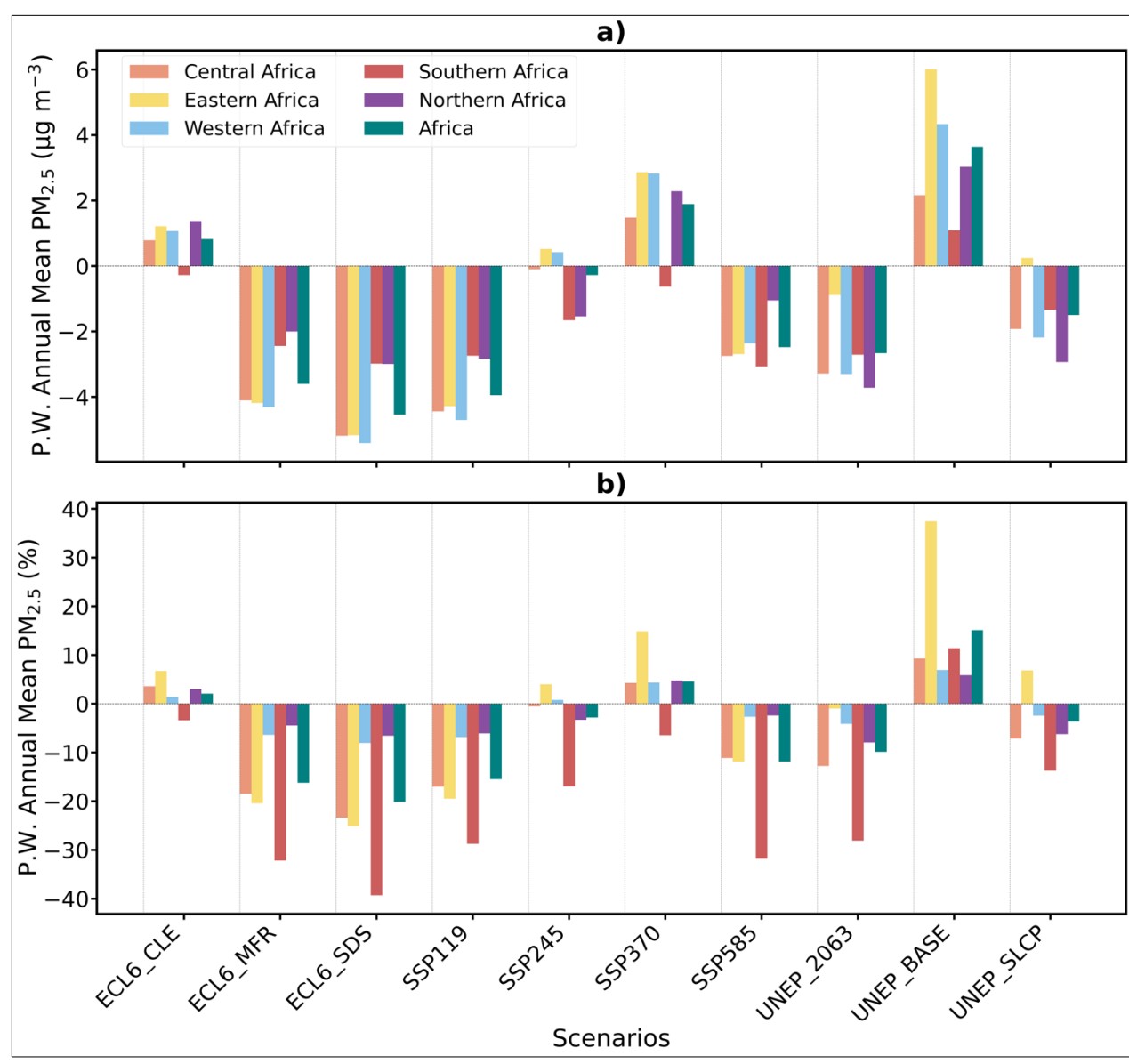

**Figure 6:** Regionally-averaged annual-mean population-weighted PM$_{2.5}$ concentrations in 2050 compared to the baseline. a) Absolute PM$_{2.5}$ changes and b) relative PM$_{2.5}$ changes.

**3.4 Excess Deaths Attributable to PM$_{2.5}$**

Building on the exposure results, we estimated the projected excess deaths per year due to PM$_{2.5}$ for baseline and 2050 emissions for each scenario. **Figure 7** shows the excess deaths due to PM$_{2.5}$ for the baseline and 2050 emissions for each region and scenario. There are large increases in almost all cases primarily due to the projected population increase (**Figure 8**). Africa's population is expected to double by 2050 under SSP370, growing from 1.1 billion to 2.3 billion, with Western Africa's population reaching 0.9 billion, followed by Eastern Africa with 0.7 billion, Southern Africa with 0.26 billion, Central Africa with 0.24 billion, and Northern Africa with

0.22 billion, as illustrated in **Figure S11**. Western Africa is projected to have the largest excess
deaths per year from ambient $PM_{2.5}$ across all scenarios, followed by Eastern Africa, Northern
Africa, Central Africa, and Southern Africa. By 2050, excess deaths in Western Africa are
projected to increase by more than 2.5 times under the highest emissions SSP370 and UNEP
Baseline scenarios, compared to 0.18 (95 CI: 0.12-0.24) million in the baseline period. Even under
the lower emission SSP119 scenario, excess deaths are projected to increase by more than 1.5
times, driven by significant population growth combined with aging (**Figure S11**). Southern Africa
consistently has the lowest projected excess deaths per year across all scenarios, with estimates
remaining under 0.05 million by 2050, although this represents up to a twofold increase compared
to the baseline.

Aging and population growth are both key aspects of the demographic transition, each
contributing to increased excess deaths. Aging increases excess deaths from age-related
diseases, even in scenarios with modest population growth. In parallel, overall population growth
amplifies excess deaths, particularly in regions with high fertility rates and younger populations,
such as Western Africa and Eastern Africa. In Northern Africa, while population growth is
projected to be slower, the aging population resulting from demographic shifts is projected to lead
to a substantial number of excess deaths, reaching 300,000 under SSP119 and SSP585, and
exceeding 200,000 in other scenarios (**Figure 8**). Southern Africa is projected to experience the
lowest number of excess deaths attributable to aging and population growth, except under
SSP119 and SSP585, where demographic transition-related excess deaths increase to about
20,000.

As illustrated in **Figure 8**, Southern Africa is projected to benefit significantly from the
pollution transition, with reductions in pollution-related deaths approaching 10,000 under
scenarios such as SSP119, SSP585, ECL6 SDS, and ECL6 MFR, primarily driven by
improvements in industrial emission controls. In other regions, the reductions are more modest,
reflecting varying levels of emissions control and air quality improvements. Most regions are
projected to benefit from the epidemiological transition, characterized by a shift from infectious
diseases to non-communicable diseases (NCDs). Excess deaths associated with this transition
are projected to decline across most regions and scenarios, except Central Africa under SSP119
and SSP585, where increases are projected. The most substantial reductions are projected under
SSP119 and SSP585, with Western Africa, Southern Africa, and Northern Africa each expected
to experience declines of up to about 100,000 excess deaths.

The wide range across emission scenarios for the future is evident in the projected excess
death estimates for all African regions, with projected excess deaths varying by a factor of 1.8
times in Eastern and Central Africa, 1.5 times in Western Africa, and up to 2.6 times in Southern
Africa. Due to large increases in population size in all African regions, $PM_{2.5}$-attributed mortality
is projected to increase in all scenarios between the baseline and near future, even in scenarios
projecting modest $PM_{2.5}$ decreases (ECL6 SDS, ECL6 MFR, SSP119, SSP585, and UNEP 2063)
and strong decreases in baseline disease rates (depicted by prevalent negative blue bars for
epidemiological transitions in **Figure 8**).

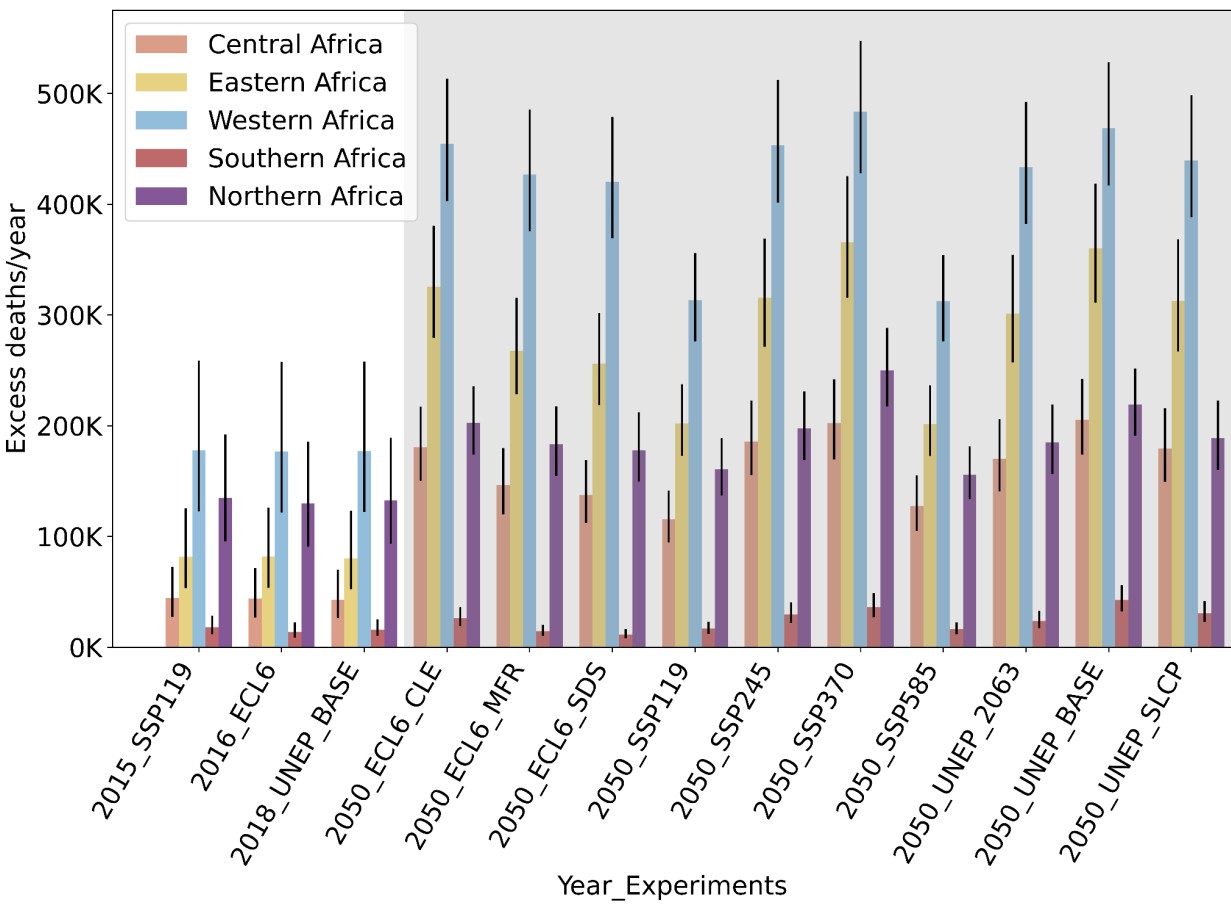

**Figure 7:** Excess deaths due to $PM_{2.5}$ calculated for baseline and 2050 emissions for each region and scenario. The black bars represent error bars, showing the 95% confidence interval in the excess death estimates for each region and scenario, reflecting uncertainty propagated from the exposure-response function (MR-BRT).

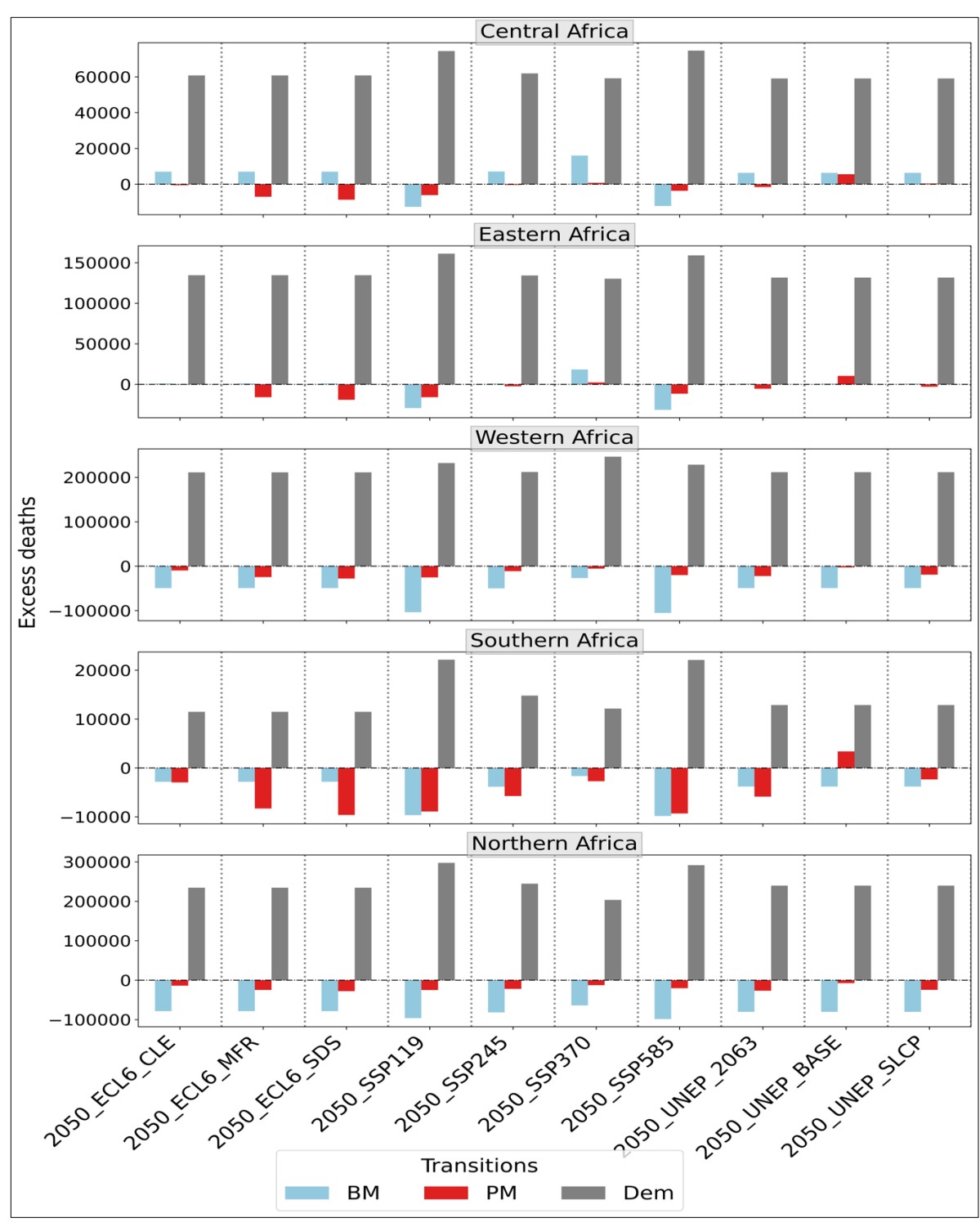

**Figure 8:** Expected change in deaths in 2050 relative to the baseline, due to epidemiological transitions (BM), pollution transitions (PM) and demographic transitions (Dem).

### 3.5 Anthropogenic aerosol-induced radiative forcing

In **Figure 9**, we show the net radiative forcing due to changes in anthropogenic aerosol emissions between the baseline and 2050, depending on the emission scenario. This includes the radiative forcing due to aerosol-radiation interactions (RFari, shown in **Figure S14**), estimated using offline radiative transfer calculations with a multi-stream model based on the discrete ordinate method (Myhre et al., 2013; Stamnes et al., 1988) and from aerosol-cloud interactions (RFaci, shown in **Figure S15**), estimated using a parameterization based on Quaas et al. (2006). The aerosol-cloud interaction estimate includes only the first indirect effect (cloud albedo effect), not the second (cloud lifetime effect). Averaged over the entire continent, the RFari in 2050 relative to the baseline is negative in SSP119, ECL6 SDS, and the UNEP scenarios, and positive in the remaining scenarios, reflecting a small cooling or warming effect depending on the scenario. The RFaci is negative in the UNEP Baseline and UNEP SLCP and positive in the SSPs, UNEP Agenda 2063, and ECL6 scenarios.

A net positive aerosol forcing in 2050 relative to the baseline is projected in all but two scenarios, with continent-wide mean values ranging from 0.03 W m$^{-2}$ in SSP119 to 0.27 W m$^{-2}$ in SSP585. Overall, we find a small warming effect due to reduced aerosol concentrations, with cooling in the two scenarios where emissions increase. Regional patterns of aerosol forcing reveal negative and positive values in certain areas, depending on the scenario, and align closely with AOD changes (**Figure S13**). This shows the dominant influence of changes in scattering aerosols in the scenarios. Although local emission changes play the most important role, there could be influences from emission changes in surrounding regions via long-range transport. Southern Africa experiences anthropogenic aerosol-induced warming in all scenarios except UNEP BASE and UNEP SLCP. This warming reflects stricter policies and subsequent stronger emissions reductions (**Figure 2**). Since Changes in scattering aerosol emissions dominate, the result is a positive radiative forcing.

The regional pattern of radiative forcing reflects not only the overall stringency of air pollution control in the scenarios but also subsequent changes in the relative importance of different aerosol emissions. The less stringent scenarios (UNEP BASE and UNEP SLCP) result in negative aerosol radiative forcing across Africa due to higher concentrations of particularly scattering aerosol. In particular, the UNEP SLCP scenario targets BC, resulting in flat or decreasing BC emission profiles, depending on sub-region, from the baseline to near future, but has nearly the same SO$_2$ increase as in UNEP BASE. Hence, the negative radiative forcing is nearly as strong in UNEP SLCP as in UNEP BASE. Although the cooling by SO$_2$ offsets a portion of the warming caused by greenhouse gases, the adverse health effects in UNEP BASE and UNEP SLCP scenarios due to poor air quality cannot be ignored. As expected, the more stringent scenarios (ECL6 SDS and ECL6 MFR) result in stronger positive aerosol-induced radiative forcing, thus contributing to warming but with reduced health risks due to improved air quality (Chalmers et al., 2012; Lund et al., 2019; Westervelt et al., 2017; Zhang et al., 2025).

Net RF

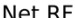

**Figure 9:** Net aerosol-induced radiative forcing estimated due to changes in emissions between the baseline and 2050.

## 4. Discussion

Emissions of aerosol species across Africa show substantial variability in existing inventories and over time, influenced by both regional factors and scenario-specific dynamics. The scenarios are designed to have large spread, reflecting uncertainties in how African nations will develop and lack of detailed assumptions in some cases. But there is also significant spread between the scenarios that assume weak or strong air pollution mitigation. The recent emission inventories differ and often lack harmonization with scenario data; although CEDS has now been extended to 2022, there is generally a delay in updating inventories and aligning baselines, highlighting a need for improved data to enhance inventory construction and reduce uncertainties.

For instance, BC emissions show variability throughout the historical period, but $SO_2$ emissions
exhibit more consistency between inventories, particularly in recent years, where 'real-world'
emissions track the high projections. These uncertainties in emissions can propagate through
climate and health impact assessments, influencing both the magnitude and spatial distribution
of projected outcomes. While the general patterns identified in our study, such as the dominant
role of population growth and the regional distribution of excess mortality, remain consistent
across scenarios, the wide range of possible emission trajectories introduces uncertainty in the
quantitative estimates of future impacts. Therefore, conclusions regarding absolute outcomes
should be interpreted with caution, whereas relative differences across scenarios and regions are
more robust. The large spread and regional heterogeneity in projected air pollution emissions are
consistent with findings from previous studies (Keita et al., 2021; Shindell et al., 2022). For
example, Shindell et al. (2022) highlight that Southern Africa is projected to have the highest $SO_2$
emissions from power plants in the industrial sector, while Western and Eastern Africa are
expected to see the largest BC emissions, primarily from the residential sector, under the high-
emission SSP370 scenario, by mid-century. Additionally, Bonjour et al. (2013) and Chowdhury et
al. (2023) noted that over 90 % of households in sub-Saharan Africa rely on solid fuels for cooking
and domestic activities, further emphasizing the dominant role of the residential sector in BC
emissions across all scenarios for the region.

Sub-Saharan Africa is undergoing two major simultaneous transformations: a shift from
traditional to modern sources of environmental pollution and a significant change in the disease
burden, moving from communicable to non-communicable diseases (Bigna and Noubiap, 2019).
Despite these shifts, household air pollution (HAP) continues to contribute to over 70 % of
anthropogenic $PM_{2.5}$ emissions in the region (McDuffie et al., 2020). Although HAP is gradually
decreasing in certain areas as households transition to cleaner alternatives (Bescond et al.,
2019), the pace is inconsistent across the continent. Notably, Northern Africa stands out as the
sub-region with the least emissions from the residential sector, attributed to greater adoption of
LPG for household activities.

Our findings on population-weighted $PM_{2.5}$ exposure align with those of Shindell et al.
(2022), who observed that increases in $PM_{2.5}$ exposure under the SSP370 scenario are notably
large for East, Northern, and West Africa, while being relatively modest in Central and Southern
Africa. The variation in national-level $PM_{2.5}$ exposure across the African region is also evident,
with significant increases observed in countries such as Nigeria, Egypt, Ethiopia, Uganda,
Rwanda, and Burundi under the SSP370 scenario. Shindell et al. (2022) study was conducted
using only the SSPs, with emphasis on the SSP119 and SSP370 scenarios, whereas our study
expanded upon this by including a broader set of scenarios, including SSPs, ECLIPSE, and UNEP
scenarios, to explore a diverse set of future possibilities under plausible conditions. Similarly,
Wells et al. (2024) study using only the SSPs identified elevated $PM_{2.5}$ exposure levels across
tropical regions of Africa. Furthermore, Chowdhury et al. (2020, 2022) found that Nigeria exhibits
the highest $PM_{2.5}$ exposure levels in West Africa.

Shindell et al. (2022) found that, under the SSP119 scenario, annual premature deaths
due to $PM_{2.5}$ are projected to decrease by approximately 515,000 by 2050, compared to the
SSP370 scenario, with reductions of 100,000, 175,000, 55,000, 140,000, and 45,000 in Northern,
West, Central, East, and Southern Africa, respectively. Wells et al. (2024) estimate that if Africa
follows the high-emission SSP370 pathway instead of the low-emission SSP119 pathway, there
could be approximately 150,000 additional deaths per year from $PM_{2.5}$ exposure. The

implementation of the strictest emission reductions could have a substantial positive impact on
public health outcomes but would still result in several hundred thousand excess deaths in the most populated regions. Pai et al. (2022) found that, even under an extreme abatement scenario with no anthropogenic emissions, more than half of the world's population would still experience annual $PM_{2.5}$ exposures above the 5 μg m$^{-3}$ guideline, including over 70 % of the African population and more than 60 % of the Asian population. This is largely due to natural sources
such as fires and dust, which aligns with our findings.

Several global studies have explored future aerosol-induced radiative forcing and climate impacts using SSPs (Lund et al., 2019) and Representative Concentration Pathway (RCP) projections (Chalmers et al., 2012; Gillett and Von Salzen, 2013; Westervelt et al., 2015). While present-day and future radiative forcing estimates vary across studies, a consistent global finding
is the significant weakening of aerosol radiative forcing by 2100 across all scenarios (Lund et al., 2019). However, Lund et al. (2019) also highlighted substantial regional differences, particularly in South Asia and Africa, where the magnitude and even the direction of forcing change vary depending on the scenario. Chalmers et al. (2012) and Gillett and Von Salzen (2013) also investigated whether the rapid decline in aerosol emissions could lead to near-term warming. For
instance, Chalmers et al. (2012) observed higher near-term warming in RCP2.6 compared to RCP4.5, despite lower greenhouse gas forcing in the former, highlighting the role of decreasing aerosol emissions. In contrast, Gillett and Von Salzen (2013) found no evidence of accelerated near-term warming linked to reduced aerosol emissions, underscoring the variability in model responses to aerosol changes. This inter-model variability is further highlighted by results from
the Coupled Model Intercomparison Project Phase 6 (CMIP6), where (Thornhill et al., 2021) show large differences in aerosol effective radiative forcing across models, reflecting substantial uncertainty in how aerosol-climate interactions are represented. While these previous studies primarily focused on global trends, and the dynamics of aerosol-induced radiative forcing may vary regionally, particularly in Africa, they provide crucial insights into how aerosol radiative
forcing evolves over time, particularly in response to changes in aerosol emissions.

Although our study does not consider the impacts of aerosols on atmospheric circulation and precipitation, prior research highlights their significant influence. For instance, Myhre et al. (2017) examined the implications of BC absorbing short-wave radiation, highlighting its potential to warm the atmosphere and cause contrasting effects on circulation patterns and regional
precipitation. Shindell et al. (2023) highlighted that Africa could significantly mitigate rainfall declines by implementing the ECL6 SDS scenario, which focuses on transitioning away from fossil fuels and minimizing food waste. Even though these changes may lead to modest near-term warming due to reduced aerosols, they offer long-term climate benefits.

Shindell et al. (2023) also noted that under the high-emission UNEP Baseline scenario,
significant drought is projected in the Sahel, whereas implementing the Agenda 2063 scenario could prevent this drying and potentially lead to a slight increase in precipitation. Previous studies have shown that local reductions in African anthropogenic aerosol emissions significantly influence the West African Monsoon (WAM) and Sahel summer precipitation (Hirasawa et al., 2022; Shindell et al., 2023; Wells et al., 2023; Westervelt et al., 2018). However, the considerable
uncertainty in aerosol emissions over northern Africa continues to contribute to the challenges in projecting Sahel precipitation changes in the near future (Monerie et al., 2023; Shindell et al., 2023; Toolan et al., 2024).

Our approach does not capture climate-driven changes in $PM_{2.5}$, which are expected to be smaller than emission-driven changes (Westervelt et al., 2016). Additionally, while we assess uncertainty from emissions, using a single model means we do not explore uncertainties in model response, such as differences in simulated concentrations and forcing.

## 5. Conclusion

In this paper, we examined the wide range and regional heterogeneity in projected air pollution emissions in Africa. We explored sub-regional and sectoral contributions to air pollution emissions in ECLIPSE, SSPs and UNEP scenarios identifying sectoral changes that influence air quality across Africa's regions. We estimated the difference in simulated population-weighted $PM_{2.5}$ exposure as well as continental and regional annual mean changes in $PM_{2.5}$ concentrations in 2050 relative to the baseline. Furthermore, we estimated the excess mortality associated with $PM_{2.5}$ exposure based on the baseline and projected 2050 emissions for each scenario. Lastly, we calculated radiative forcing due to changes in emissions between 2050 and baseline.

Using the ECLIPSE, SSPs, and UNEP emission projections, we find substantial sub-regional differences in BC, $SO_2$, OC, and $NO_x$ emissions across Africa. Emissions vary substantially depending on the scenario and region, with high emissions of BC, OC, and $NO_x$ in Western Africa, Eastern Africa, and Central Africa, and high emissions of $SO_2$ in Southern Africa. Key sectoral contributors to air pollution vary by region, with the residential, industrial, transportation, and energy sectors playing dominant roles depending on the region and scenario. In the baseline and by 2050, the residential sector will remain the largest source of BC emissions in sub-Saharan Africa, while in Northern Africa, the transportation sector leads. Sectoral emission distributions differ across SSPs, ECL6, and UNEP scenarios due to varying assumptions about regulations, technology, and economic development. Better and more spatially resolved assumptions about policy, technology, and economic development are needed for projections on the continent, especially for the upcoming IPCC AR7.

Regionally-averaged annual-mean $PM_{2.5}$ concentrations exhibit distinct trends across different African regions, with variations driven by the stringency of emission scenarios. We find that regionally-averaged annual-mean population-weighted $PM_{2.5}$ concentrations are projected to increase by up to 6 µg m$^{-3}$ (37 %, SD ± 2.7 µg m$^{-3}$) under the UNEP Baseline, SSP370, and ECL6 CLE scenarios by 2050, when averaged across the African continent. Conversely, decreases of up to 5 µg m$^{-3}$ (8 %, SD ± 2.5 µg m$^{-3}$) in Western Africa and 3 µg m$^{-3}$ (39 %, SD ± 1.4 µg m$^{-3}$) in Southern Africa are projected under the ECL6 SDS and MFR scenarios. Western, Eastern, and Northern Africa are projected to experience the highest exposure levels under the UNEP Baseline, SSP370, and ECL6 CLE scenarios, driven by increased emissions from the residential, industrial, transportation, and energy sectors. In contrast, Southern Africa is projected to see declines in annual $PM_{2.5}$ levels under the SSP370 and ECL6 CLE scenarios, with reductions of 0.6 µg m$^{-3}$ (6 %) and 0.3 µg m$^{-3}$ (3 %), respectively, primarily due to decreased emissions from the energy and industrial sectors. By 2050, annual-mean $PM_{2.5}$ concentrations for Africa differ by up to a factor of 2 between the highest emission scenario (UNEP Baseline) and the lowest (SSP119) when averaged across the continent, further highlighting a substantial spread in future emission projections.

Substantial increases in excess deaths due to $PM_{2.5}$ are projected in almost all scenarios, largely driven by significant population growth. We find that by 2050, Western Africa is projected

to experience the highest increase in excess deaths across all scenarios, with an increase of over
2.5 times compared to the baseline under the SSP370 and UNEP Baseline scenarios, driven
primarily by population growth and aging. Eastern, Northern, Central, and Southern Africa follow
in terms of projected excess deaths, but Southern Africa is projected to have the smallest
increase, remaining below 0.05 million. While $PM_{2.5}$ reductions may slightly decrease mortality in
some areas, population growth will still lead to hundreds of thousands of excess deaths,
underscoring the need for significant $PM_{2.5}$ reductions. Epidemiological and pollution transitions
are projected to tend to reduce excess deaths in all regions, particularly in Western, Southern,
and Northern Africa, with notable reductions in Southern Africa under certain scenarios. However,
the overall effect is primarily driven by population growth.

We find a net positive aerosol-induced forcing across Africa in all scenarios, by 2050,
except UNEP BASE and UNEP SLCP scenarios, with values ranging from 0.03 W $m^{-2}$ in SSP119
to 0.27 W $m^{-2}$ in SSP585, driven mainly by changes in scattering aerosols. Regional patterns
show both positive and negative aerosol forcing depending on the scenario, with Southern Africa
experiencing warming in all but two scenarios (UNEP BASE and UNEP SLCP) due to reductions
in industrial emissions. The less stringent scenarios (UNEP BASE and UNEP SLCP) lead to
negative aerosol radiative forcing across Africa, primarily due to higher scattering aerosol
concentrations, with UNEP SLCP focusing on eliminating BC emissions but with no changes to
$SO_2$ emissions. Further work is needed to quantify the associated climate implications and risks.

This work provides profound insights into Africa's complex and diverse emissions
landscape and lays the groundwork for future research and policy interventions. Understanding
the drivers of change and the spread in scenarios is critical for addressing future emissions and
their impacts. Differences in emissions projections highlight the need for robust analysis of socio-
economic, regulatory, and technological pathways. Additionally, improving present-day data is
essential to better constrain the baseline for scenarios, ensuring that models accurately reflect
current emissions and activity levels. Accurate activity data and harmonization are especially
essential for upcoming assessments aimed at informing policy makers, such as the various
elements of the 7th Assessment Report of the Intergovernmental Panel on Climate Change, which
will rely on these baselines to effectively consider the implications of future climate and air quality
policies.

**Code Availability**
The OsloCTM3 model is available from GitHub - NordicESMhub/OsloCTM3: Oslo CTM3 – A
global chemical transport model (details about version used in the present study will be provided
upon request to the corresponding author). The data analysis was conducted using Bash scripting
and Python programming languages.

**Data Availability**
The OsloCTM3 model outputs are available here:
https://figshare.com/account/items/28030610/edit. NASA data is accessible through Giovanni,
and U.S. Embassy data can be found on AirNow. Satellite-derived PM2.5 data is available via the
Satellite-derived PM2.5 Archive | Atmospheric Composition Analysis Group | Washington
University in St. Louis (wustl.edu. The SSP scenarios are provided by IIASA, and baseline
mortality data can be accessed through the Global Burden of Disease Results Tool.

**Author Contributions**

MTL, BHS, DMW, SC, GM, ANJ, and JAA conceptualized and designed the study. MTL performed the OsloCTM3 simulations; ANJ processed and prepared emissions; GM performed the radiative forcing calculations; SC performed the health impact assessment calculations; JAA performed data analysis and led the writing; All authors reviewed and edited the paper.

**Competing Interests**

The authors declare no conflict of interest.

**Acknowledgements**

The authors acknowledge funding from the Research Council of Norway. The authors acknowledge the UNINETT Sigma2 – the National Infrastructure for High-Performance Computing and Data Storage in Norway – resources (grant no. NN9188K). We also acknowledge support by the Center for Advanced Study in Oslo, Norway which funded and hosted the HETCLIF centre during the academic year of 2023/24.

This work was also supported by the National Science Foundation Office of International Science and Engineering (OISE) Award Number 2020677.

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
