# Peer review of "An uncertain future for the climate and health impacts of anthropogenic aerosols in Africa"

_EGUsphere, 2025_

## Referee Comment (RC1)

General comments

This paper addresses an important question regarding the uncertainty – and associated causes – of emissions scenarios over Africa, and the consequent uncertainty in health and climate responses. The method is introduced clearly, and the results are presented in a sensible manner overall, with appropriate conclusions drawn. Overall, this paper is a nice contribution to the field and will aid in the understanding of future plausible emissions trajectories over Africa.

However, the ease of interpretation of the study's results could be decently improved with some modifications to the paper. Below are some general overview comments, with more specific line-by-line comments underneath (some of which relate to the overview comments). On the basis of these suggested changes to the presentation of the study, and in the context of the sensible methodology and important results, I recommend this study be accepted subject to minor revisions.

Some sections of the results would benefit from expansion and clarification (especially the health results), potentially at the expense of parts of the emissions results, if space is short. Sections 3.1 and 3.2 should be condensed, with key narratives drawn out, and presented in a way which is easier to follow and absorb. Section 3.2 is currently better in this regard, but could still benefit from improvement. This could involve reducing the amount of information presented. On the contrary, the health results in section 3.4 could be expanded, as they only contain 2 paragraphs and no in-text reference to figure 7. The relative effect of aging vs overall population growth isn't discussed in this section, but is of interest in interpreting the results.

Several of the figures could also be substantially improved in their presentation. Figures 3 and 4 are somewhat squashed, and the maps in figure 5 should be enlarged. There are specific comments on figures below.

Line-by-line comments

Introduction

L10: the abstract doesn't mention the fact that baseline emissions vary in their sectoral split, which is an important result.

L39: is this the sensitivity to local emissions reductions? Or remote?

L42: it's worth mentioning the semi-direct burnoff effect too.

L46: it's probably worth referring to IPCC AR6 on this too – as the definitive and more recent summary; or even the Indicators of Global Climate Change studies for more recent, and IPCC-consistent, analysis https://essd.copernicus.org/articles/16/2625/2024/.

L52: "more" recent than what? The prior references? I think it would be clearer to drop this word. Also perhaps quickly clarify this is the global burden.

L60-63: this might imply these changes are similar between scenarios – so perhaps worth caveating here to note the future uncertainty in this (which you of course go on to explore).

L74: I think it's worth explicitly noting the increased understanding in both history (from inventories) and future plausible trends (from scenarios).

L75: I think "anticipated development" should be rephrased – "accelerating development"? I think a brief noting of the wide spatial variation across the continent in any concept of development would be useful too.

L82: it might look unwieldy but you should repeat "SSP" within the scenario names e.g. SSP3-7.0, since these are the exact names.

L81-90: this paragraph is a bit hard to parse; I think you could drop the specifics of the 10 emissions pathways as you detail them in the methods, and it isn't essential at this point to know what they are. Similarly with details of the variables for evaluation. This paragraph should just be a brief overview of the method; no need for this much specificity.

L95: more detailed representation compared to what?

L98: "observed and satellite data" might imply the satellite data isn't observed? Just a quick rephrasing needed for clarity.

L104: "between 2050 and 2015-2018" – switch these around I think for ease of reading; should be "from past to future".

Methods

L108: worth defining WHO, and also for reproducibility a brief explanation of those minor adjustments, in the supplement.

L114: not sure it's worth stating your overall method again here.

Figure 1: I think the SSA is referring to the country boundaries being a different colour, but this isn't obvious. It would be best to just have the border round the actual region be distinct, and not the internal borders. Also clarify somewhere that sub-Saharan = West + East + Central + South (if I've understood that right).

L122: see above on SSP scenario names. It would probably be better to list the scenarios in 3 bullet points, maybe 1 for each source ie SSPs, ECLIPSE, UNEP? Use specific names – perhaps shorthand ones – and use these throughout the paper.

L127: what is the resolution of these new emissions? How does this vary across the emissions inventories?

L146: "[to provide emissions] for 2018 and 2050" – would suggest this, for clarity

L148: "be developed, in contrast" – this comma should be a full stop or semi colon.

L150: ", like advances" – would replace "like", maybe "such as"?

L160: would use semi-colons to separate this list, for clarity.

L164: "SSPs" to "SSP"

L165: do you not need a paragraph on the ECLIPSE emissions? Or is the info in the first paragraph of this section sufficient (I think it could be expanded to explain the motivation of

these emissions)? In any case, I would put this info in separate paragraphs, in the order in which you introduce the scenarios at the start of this section, to aid the reader. I think you need to be clearer on which species are included in each inventory – do they all give the same?

L168: Don't need "The" at the start.

L178: would drop the word "recently"; as long as it's the documentation for the model you used, this doesn't matter, and 2018 is debatable as recent in a fast-moving field!

L180: can you say roughly which fields are initialised in the model?

L180: I think this should be a new section, on "Experiments" – since you have information on the simulations, rather than just the model itself. Readers looking for the experimental setup won't necessarily expect it to be in the model documentation section.

L191: technically 2015 is the first "future" year for the SSPs, right? So maybe drop the bit in brackets.

L194: this 2% is not just due to the different time period, but also the inventories themselves, right? This strengthens your justification for using different baseline years.

L195: info on this should just be in Section 2.4.

L200: "[and] ammonium". Is the fine-mode defined with an upper level at 2.5um? i.e. giving exactly PM2.5? This is important to clarify.

L203: need "-" between "population weighted"

L204: I would drop the 2015, 2016, 2018 as you have already described this; just refer to something like "respective baselines" or even just "baselines".

L206: population isn't an output from ISIMIP "simulations"; it's an input – so just clarify that.

L207: minor point but just say 1 degree resolution – you didn't specify lat and lon in the 0.5 deg population above.

L209: I'm not sure what this sentence means – the model could be ran in 1 degree resolution? I'm not sure how that shows the downscaling works well?

L212: how did you assign gridcells to countries, when borders will cut across gridcells? Just need to clarify the method used.

L214: "SSP population" not "SSPs.. ". This approach seems sensible, as you have the population data for SSPs but presumably there aren't the same for the others, but it's worth briefly exploring the sensitivity to this – since this is an inconsistency in the other scenarios. i.e. do you get the same overall results if you just use SSP2 for the SSPs as well?

L243: what does level 3 mean? If not relevant, can leave out.

L244: see above for consistency in resolution formatting.

L251: not sure about using "our" here – maybe clarify the research group? No strong opinion on this though.

L252: reference for the recent GBD study?

How do you deal with the uncertainty in this function? Do you take draws? What about the low concentration threshold below which there is no impact? There also should be something brief on how this function was made, and its applicability for Africa.

L262: this paragraph is a bit harder to follow; I would define the terms first and then expand on them. The use of a 25km2 grid needs more justification – where does this come from? Is it meaningful to downscale the CTM results to this extent, or would it be better to upscale?

L275: 5km2 is a typo?

L277: this use of SSP2 was mentioned above, without this justification; this should be included in that earlier part, with a reference.

Results

L283: why "sub-Saharan"? This is the same as all the non-north Africa regions I guess? I think Figure 1 could be improved to be more clear on this. There is a lot of variation in North Africa too, so I think you undersell yourself by focusing on a subset of the continent here.

Figure 2: Is "ECL6" a duplication? It's not distinct from others, and there are 11 scenarios in the legend but you mention using 10 above. SAF = Sub-Saharan Africa? That needs defining somewhere; if you are going to use that then you can include it in figure 1. But maybe it's worth just writing it out in full like the other regions. Note in the caption that the y scales vary between regions. It's interesting that CEDSv21 is lower than 17, right? Also that formatting "CEDSv21" is different from the main text.

L294: SSP245 and SSP585? Also check formatting – it was SSP2-4.5 earlier. Just be consistent.

L300: I don't think Wells et al., 2024 focused on SSP585 – they mention SO2 emissions in SSP119 being driven by industry, which is relevant to your next sentence, but this reference to BC in SSP585 is worth checking. That next sentence isn't quite right either – SO2 emissions increase to mid-century in SAF in SSP119 and SSP585, before decreasing; you might be saying this already but worth clarifying for sure.

L308: would simplify by saying "the ECLIPSE scenarios"; generally, this section could do with a bit of tightening up just to simplify and aid the reader. Currently it's a bit overwhelming and hard to take in all the detail; the reader would take away more if there was less info. Figure out the key trends you want to highlight, and consider dropping some others.

L310: to 2050 right?

L318: needs a reference here; also worth noting this dynamic (reducing warming SLCPs but not SO2) earlier when you introduce the scenarios.

L337: "the SSP370" – needs rephrasing. Again, lots of information in this bit which the reader may struggle to absorb (speaking from experience!).

L361: could do with explaining exactly why it's important for accurate sectoral information (i.e. IAM consistency?).

Figures 3+4: these are a nice way of showing this, but these need some work. The figures are currently squashed making them harder to read. Maybe you could have one comparative

stacked bar chart for each region instead of the 3 pie charts? Like the supplement. But maybe that won't work. List the "increasing scenarios" in the caption, not just the main text.

L376: "the SSP585"

Section 3.2 has some really nice material, and is clearer than the previous section. E.g. the last paragraph, L367-9, L388-90 are clear; it's great when you can put these differences in their broader context. This still needs some work though to condense out the key narratives you want to get across. In general you should briefly say what the supplementary figures show as well as the conclusions drawn from them. You slip into discussing "the SSPs" when you are only looking at SSP370 in this bit, right? Need to be careful on that.

L437: this sounds like you're referring to prior analysis; ned to be clear these are your results.

L439: define MAE, RMSE. Maybe make the point here that there is some disagreement between the observations? And set your results in that context.

Figure 5: this needs tidying up too; the maps are small and the plot could be better formatted. Maybe use a discrete scale to make it easier to read roughly what each value is.

Again, you should introduce the figures in the text with roughly what they show, so the reader doesn't have to jump to the figure straight away to get the picture.

Figure 6: you should use the same regional colour scheme as in the maps for these lines; I think you've used close colours but they're not exactly the same. Maybe it would be worth adding a second panel showing the percentage changes, as these are not the same as the magnitude ones and also tell an interesting story?

L468: "exposure [increase]"?

L483: you need to show that this is the case – see suggestion for figure 6 above.

L486: this is nice – relating your results to the underlying drivers.

L492: you use "MFR" here instead of "ECL6 MFR" – it's fine to have shorthand (and maybe this would be a good thing to use throughout) but need to be consistent. If you use shorthand names, then define them when you introduce the scenarios – see the suggested in the intro about bullet points for these scenarios.

L494: don't use PM as shorthand for PM2.5; you need to write it out in full if this is the metric we are looking at.

L511: "s partially due" -> primarily due, right?

L512: "Western African" - Western Africa's? also, need to be clear this is SSP370 (right?), as there is wide variation in the figure you refer to.

Section 3.4: this section was a bit of an anticlimax – this is one of your main results, and it's only 2 paragraphs! Figure S13 is really interesting, showcasing the different drivers of these changes – and that these are fairly consistent across regions, with some important scenario and regional differences. Figure 7 is great (though the colours don't seem quite the same as figure 6, and the order is different, which should just be fixed), but you don't refer to it in the text at all! Though you do quote data from it.

L531: I don't think this conclusion about baseline concentrations follows, unless you make an argument based on the functional form of the response function? The population increase would drive increases regardless of the baseline, all else being equal.

Figure 7: are these error bars from the response function?

"3.5 Anthropogenic aerosol-inducing radiative forcing" – aerosol-induced?

L542: "s between 2050 and 2015, 2016 or 2018," – again, just say baseline, and go from present to future.

L544: should note again here that this is only the 1st cloud effect, not the 2nd (lifetime) one. also just name again the method/model used.

L551: clarify regional-mean is SAF.

L559: "emissions of changes" -> "changes of emissions"?

L552: before you go into detail here, just state the overall result: you generally find a small warming effect due to reduced concentrations, with cooling from the increase in 2 scenarios.

L584: "CEDS is not updated to 2022" – not sure when this sentence was written but CEDS is available to 2022 now; maybe just note a general delay in inventories?

L636: "significant weakening of aerosol radiative forcing by 2100 across all scenarios" – is this true spatially, e.g. in Africa? Or just globally?

L640: you should bring in CMIP6 studies on aerosol forcing e.g. https://acp.copernicus.org/articles/21/853/2021/ - to display the inter-model variability.

L649: "found that BC absorbs short-wave radiation," – careful with your phrasing when it comes to describing other studies; they didn't show this effect (ie for the first time); they look at the implications of this effect.

It's worth noting in the discussion that your method will miss climate driven PM2.5 changes, though these shouldn't be very large. Also you should note that while you explore uncertainty in emissions, your use of one model means the uncertainty in model response – concentrations and forcing - is not explored.

L709: this is the first time you've mentioned aging as a cause of this increase – how important is this cf the general population increase? If it's important, then your earlier presentation of results as being driven by population change should be clarified to note this is not just a numerical population increase, but also the aging effect.

L714: "projected to reduce excess deaths" – should be clear this is just the tendency of these effects, not the overall effect (which is driven by population)

---

## Referee Comment (RC2)

**General comments:**

This manuscript presents a valuable assessment of future aerosol emission projections for sub-Saharan Africa, highlighting the considerable diversity among scenarios and their implications for air pollution, health, and climate in the future. Overall, the manuscript is well-structured and presents interesting and much-needed results that contribute to understanding future emissions in Africa and their potential health impacts.

However, there are several areas that could be improved. The introduction feels somewhat disjointed, and although one of the main focuses of the article is the impact of air pollution on health, this topic is only briefly addressed. My primary recommendation is to expand on this aspect within the introduction and improve the overall narrative flow.

In addition, certain methodological details could benefit from further clarification. For example, in Section 2.2, where all scenarios are defined, it is at times difficult to follow which and how many scenarios are actually used. Moreover, the rationale behind the selection of scenarios should be better justified.

Some figures also lack sufficient quality—Figures 3 and 4 in particular would benefit from improved resolution and clarity.

Finally, although the manuscript acknowledges uncertainties in emission inventories, providing more detail on how these uncertainties might influence the main conclusions would offer a clearer perspective on the robustness of the findings.

Addressing these points would significantly enhance the clarity and impact of the manuscript. These comments, along with minor technical corrections, are detailed in the specific and technical comments below. Overall, this study makes a strong contribution to the field and, with some adjustments, will serve as a valuable resource for future research and policy discussions.

**Specific comments:**

**Introduction:**
As mentioned previously, the introduction appears somewhat brief and simplistic. It could benefit from improved coherence and a more fluid narrative. Moreover, it lacks sufficient information on the health impacts of air pollution, which is one of the central pillars of this study. Strengthening the connection between changes in air pollution and their health impacts would enhance the cohesion and relevance of the introduction.

Additionally, the manuscript could better clarify why this study is necessary in comparison to previous research. What are the specific knowledge gaps this study aims to address?

**2. Methods:**

**2.1 Study Domain:**

There is insufficient justification for the chosen study domain. For instance:

- **Justification for regional division:** Why were these five specific subregions selected, and how do they influence the analysis?

- **WHO region adjustments:** A brief explanation of why modifications to WHO regions were made and what those adjustments entailed would be helpful.

- **Definition of boundaries:** How were the boundaries for each subregion determined—based on geographic coordinates, political borders, or climatic characteristics?

- **References and support:** Including additional references that support the chosen classification or that have used similar regional divisions in previous studies would strengthen this section.

**2.2 Emissions:**

This section contains a lot of information on the scenarios and their characteristics. Including a summary table—perhaps in the supplementary material—would help clarify which scenarios are considered. At present, it can be difficult to grasp how many scenarios are used, what they are, and how they differ from each other.

Several scenario frameworks are presented (SSP, ECLIPSE, UNEP), but it is not clearly explained why these particular scenarios were chosen. Why these ten, and not others? A brief justification for the selection of these specific scenarios over other potential alternatives would be valuable.

Consistency in reference years: The period 2015–2018 is mentioned as the baseline, yet 2016 is used for ECLIPSE and 2018 for UNEP. What accounts for this variation? Clarification would improve the consistency of the methodology.

**2.3 OsloCTM3**

**Model Simulations and Resolution:** The OsloCTM3 model operates at a horizontal resolution of 2.25° × 2.25°. Is this resolution not rather coarse for capturing spatial details at the regional level? A brief explanation of the advantages or limitations of the OsloCTM3 compared to other chemical transport models would be helpful.

**Explanation of resolution and rescaling:** The manuscript mentions that the model runs at 2.25° × 2.25° resolution, but the data are then rescaled to 1° × 1°. It would be useful to briefly explain why this rescaling is performed and what impact it may have on the accuracy of the results.

**2.4 Observations**

**Justification for the selection of observational data:** The manuscript states that PM2.5 observations are obtained from U.S. embassies in Africa. Is there a specific reason for selecting only these data instead of including other monitoring networks? Clarifying this point would help assess whether this selection introduces any limitations regarding the spatial representativeness of the observational dataset.

**2.5 Health Impact Assessment**

**Spatial resolution:** Data are interpolated to a 25 × 25 km resolution, but final population estimates are provided at 5 × 5 km resolution. Could this discrepancy introduce biases in the estimates of excess mortality?

**Results**

Figures 7, S9, and S10 are not cited in the text. Please, add them.

As mentioned earlier, figure clarity should be improved, particularly for Figures 3 and 4. Additionally, in the health-related section, the manuscript does not discuss the relative effect of aging versus overall population growth, an important consideration for interpreting the results.

**MINOR COMMENTS:**

**Terminology consistency:**

- In some parts of the manuscript, the term *"future emissions"* is used, while in others *"2050 emissions"* appears. For greater clarity, it would be helpful to adopt consistent terminology throughout the manuscript.

- Similarly, there is inconsistency in referring to the baseline emissions as *"baseline emissions"* versus *"historical emissions"*. Please ensure consistent use of these terms across the text.

- **Cross-referencing between sections:** In the section where the model validation against observations in Africa is discussed, it would be helpful to include a cross-reference to Section 2.4 to reinforce the structure and cohesion of the manuscript.

**Line-specific suggestions:**

- **Line 249:** *"we apply"* → *"we applied"* (to maintain past tense when describing the methods used).

- **Line 252:** *"Using the cause-specific exposure-response function, we estimate excess deaths..."* → Consider revising to *"we estimated excess deaths..."* to align with the use of past tense in the rest of the methodology section.

- **Line 264:** Add *"the"* before *"concentration"*.

- **Line 264:** Add a comma before *"respectively"*.

- **Line 265:** *"Age specific RRs (Relative Risks) for IHD and stroke, are obtained using MR-BRT."* → Remove the comma after *"stroke"*: *"Age-specific RRs for IHD and stroke are obtained using MR-BRT."*

- **Line 265:** *"For LC, T2-DM, and COPD uniform RRc,d were used across all age groups among adults."* → Add an article: *"For LC, T2DM, and COPD, a uniform RRc,d was used across all age groups among adults."*

- **Line 275:** *"For the concentration data generated using SSP emissions, we apply the respective SSP projections..."* → For consistency in verb tense: *"we applied the respective SSP projections..."*

- **Line 294:** *"Notably, there are projected declines in BC emissions after mid-century under SSP245 and SSP245."* → There is a repetition of "SSP245." Based on Figure 2, it seems this should refer to "SSP585." Please verify and correct the second scenario accordingly.

- **Line 341:** *"Western Africa and Eastern"* → Suggested revision: *"Under the UNEP baseline scenario, BC, OC, and NOx emissions are projected to be largest in Western Africa and Eastern Africa, followed by Central Africa, Southern Africa, and Northern Africa, by 2050..."*

- **Line 397:** *"The SSPs also demonstrates"* → Should be *"demonstrate"* (plural subject).

- **Line 407:** *"Additionally, Under the ECL6 CLE scenario"* → The word *"Under"* should not be capitalized: *"Additionally, under the ECL6 CLE scenario..."*

- **Line 529:** *"even in scenarios projecting modest PM2.5 decreases (ECL6 SDS, SSP1-1.9, UNEP 2063, etc)"* → The use of *"etc."* is too vague for academic writing. If possible, specify the remaining scenarios or remove the phrase altogether.

- **Line 531:** *"This further highlight that there are already high PM2.5 concentrations"* → Should be *"highlights"* to match the singular subject *"This."*

- **Line 545:** *"the RFari in 2050 relative to the baseline is negative in the UNEP scenarios, SSP119, and ECL6 SDS scenarios."* → To avoid repeating *"scenarios"*, consider rephrasing: *"the RFari in 2050 relative to the baseline is negative in the UNEP, SSP119, and ECL6 SDS scenarios."*

- **Line 554:** *"This shows the dominating influence of changes in scattering aerosols in the scenarios."* → Consider replacing *"dominating"* with *"dominant"*, which is more natural in this context.

- **Line 559:** *"Since emissions of changes in scattering aerosols dominate, the result is a positive radiative forcing."* → The phrase *"emissions of changes"* sounds awkward. A clearer option would be: *"Since changes in scattering aerosol emissions dominate, the result is a positive radiative forcing."*

- **Line 561:** Add *"the"* before *"relative importance of different aerosol emissions."*

- **Line 566:** *"Hence, the negative radiative forcing is therefore nearly as strong in UNEP SLCP as in UNEP BASE."* → *"Hence"* and *"therefore"* are redundant. Consider removing one for clarity.

- **Line 570:** Typo: *"ECLE SDS"* should be corrected to *"ECL6 SDS."*

- **Line 582:** *"lack of detailed assumptions"* → Consider rephrasing to *"lack of detailed assumptions in some cases"* to make the meaning clearer, as *"lack of assumptions"* can be ambiguous.

- **Line 585:** *"there is the need for improved data"* → For smoother phrasing, revise to: *"there is a need for improved data."*

- **Line 588:** *"where "real-world' emissions"* → The closing quote is incorrect. It should read: *"where 'real-world' emissions."*

- **Line 594:** *"Bonjour et al. (2013) and Chowdhury et al. (2023), noted"* → Remove the comma after *"2023"*: *"Bonjour et al. (2013) and Chowdhury et al. (2023) noted…"*

- **Line 685:** Replace *"is"* with *"are"* in reference to *"assumptions"*, which is a plural noun.

- **Line 717:** *"by 2050 except UNEP BASE and UNEP SLCP"* → For clarity and smoother flow: *"by 2050, except in the UNEP BASE and UNEP SLCP scenarios."*

- **Line 723:** *"across Sub-Saharan Africa"* → Within a sentence, *"sub-Saharan Africa"* should be in lowercase.

- **Line 734:** *"Accurate activity data and harmonization efforts are especially essential for upcoming assessment efforts"* → Consider rewording to avoid the repetition of *"efforts"*, e.g., *"Accurate activity data and harmonization are especially essential for upcoming assessments."*

---

## Author Response (AR1)

Referee: 1
General comments
This paper addresses an important question regarding the uncertainty – and associated causes – of emissions scenarios over Africa, and the consequent uncertainty in health and climate responses. The method is introduced clearly, and the results are presented in a sensible manner overall, with appropriate conclusions drawn. Overall, this paper is a nice contribution to the field and will aid in the understanding of future plausible emissions trajectories over Africa.

However, the ease of interpretation of the study's results could be decently improved with some modifications to the paper. Below are some general overview comments, with more specific line-by-line comments underneath (some of which relate to the overview comments). On the basis of these suggested changes to the presentation of the study, and in the context of the sensible methodology and important results, I recommend this study be accepted subject to minor revisions.

Some sections of the results would benefit from expansion and clarification (especially the health results), potentially at the expense of parts of the emissions results, if space is short. Sections 3.1 and 3.2 should be condensed, with key narratives drawn out, and presented in a way which is easier to follow and absorb. Section 3.2 is currently better in this regard, but could still benefit from improvement. This could involve reducing the amount of information presented. On the contrary, the health results in section 3.4 could be expanded, as they only contain 2 paragraphs and no in-text reference to figure 7. The relative effect of aging vs overall population growth isn't discussed in this section, but is of interest in interpreting the results.

Several of the figures could also be substantially improved in their presentation. Figures 3 and 4 are somewhat squashed, and the maps in figure 5 should be enlarged. There are specific comments on figures below.

We sincerely thank the reviewer for their thoughtful and constructive feedback. We are pleased that the reviewer found the paper to be a valuable contribution with a clear methodology and sensible conclusions. In response to the reviewer's suggestions, we have made several revisions to improve the clarity and interpretability of the results. Specifically, we have condensed Sections 3.1 and 3.2 to better highlight the key narratives and expanded the discussion of health impacts in Section 3.4, including a more detailed analysis of the relative effects of aging versus overall population growth. We have also revised several figures (Figures 3, 4, and 5) to enhance readability and presentation quality. We have added an in-text reference to Figure 7 and added a significant discussion breaking down the components of the mortality changes (driven by aging, population, or PM exposure). We believe these changes have significantly strengthened the paper, and we appreciate the reviewer's guidance in helping us improve the manuscript.

Line-by-line comments
 Introduction
L10: the abstract doesn't mention the fact that baseline emissions vary in their sectoral split, which is an important result.
To address this, we added the following to the abstract: Baseline emissions also vary in their sectoral split.
L39: is this the sensitivity to local emissions reductions? Or remote?
This refers primarily to the sensitivity to local emissions reductions. We have clarified this in the manuscript.
L42: it's worth mentioning the semi-direct burnoff effect too.
To address this, we revised the sentence in the manuscript to the following: Aerosols impact surface temperature not only by directly scattering or absorbing incoming solar radiation but also indirectly by altering cloud properties, including their brightness, lifespan, and, in the case of absorbing aerosols, by reducing cloud cover through the semi-direct burnoff effect (Albrecht,

1989; Twomey and S., 1977).

L46: it's probably worth referring to IPCC AR6 on this too – as the definitive and more recent summary; or even the Indicators of Global Climate Change studies for more recent, and IPCC-consistent, analysis https://essd.copernicus.org/articles/16/2625/2024/.

We agree that referencing the IPCC AR6 provides a more definitive and recent summary, and that the Indicators of Global Climate Change study offers an important, up-to-date, and IPCC-consistent analysis. We have incorporated citations to both the IPCC AR6 and the Indicators of Global Climate Change study (Forster et al., 2024) in the manuscript to strengthen and update this section

L52: "more" recent than what? The prior references? I think it would be clearer to drop this word. Also perhaps quickly clarify this is the global burden.

Thank you for pointing that out. We agree that the use of "more" was unclear, and we have removed it for clarity. Additionally, we have added "globally" in the manuscript to specify that the 8.1 million deaths per year is referring to the global burden of $PM_{2.5}$-related mortality.

L60-63: this might imply these changes are similar between scenarios – so perhaps worth caveating here to note the future uncertainty in this (which you of course go on to explore).

To address this, we revised the sentence in the manuscript to the following: Ongoing and anticipated future changes in Africa, including rapid economic development, rising urbanization, continued dependence on traditional biomass fuels, agricultural practices, limited access to advanced pollution control technologies, infrastructure expansion, and population growth, are expected to increase exposure to air pollutants, although the magnitude and direction of these changes remain highly uncertain across future scenarios (Abera et al., 2020; Bauer et al., 2019; Chowdhury et al., 2020; Wells et al., 2024).

L74: I think it's worth explicitly noting the increased understanding in both history (from inventories) and future plausible trends (from scenarios).

To address this, we revised the sentence in the manuscript to the following: Global emission inventories, including the Community Emissions Data System (CEDS) (Hoesly et al., 2018), the Evaluating the Climate and Air Quality Impacts of Short-Lived Pollutants (ECLIPSE) inventory from the Greenhouse Gas-Air Pollution Interactions and Synergies (GAINS) model (Amann et al., 2011), and the Emissions Database for Global Atmospheric Research (EDGAR) (Crippa et al., 2018; Janssens-Maenhout et al., 2019), have improved our understanding of historical emissions trends, while global emission scenarios such as the Shared Socioeconomic Pathways (SSPs) (Riahi et al., 2017), have improved our understanding of future plausible trends.

L75: I think "anticipated development" should be rephrased –"accelerating development"? I think a brief noting of the wide spatial variation across the continent in any concept of development would be useful too.

To address this, we revised the sentence in the manuscript to the following: This knowledge gap is particularly pressing given Africa's accelerating development, substantial spatial variation in economic growth and industrialization, and the wide range of emission sources across the region.

L82: it might look unwieldy but you should repeat "SSP" within the scenario names e.g. SSP3-7.0, since these are the exact names.

Thank you for the suggestion. To address this, we added SSP in front of the scenario names in the manuscript as follows: SSP1-1.9, SSP2-4.5, SSP3-7.0, and SSP5-8.5.
To maintain consistency with the figure labels and improve clarity, we refer to the SSP scenarios using shorthand (e.g., SSP119, SSP245) throughout the manuscript, as you suggested in L122.

L81-90: this paragraph is a bit hard to parse; I think you could drop the specifics of the 10

emissions pathways as you detail them in the methods, and it isn't essential at this point to know what they are. Similarly with details of the variables for evaluation. This paragraph should just be a brief overview of the method; no need for this much specificity.

We agree that providing detailed listings of scenarios and evaluation variables at this early stage made the paragraph dense and harder to follow. we have simplified the paragraph in the manuscript to the following:  We explore the range in projections of mid-century anthropogenic air pollution levels in Africa, and the associated region-specific health impacts, resulting from 10 different pathways using the Oslo chemical transport model version 3 (OsloCTM3). We present estimates of the projected future regional radiative forcing of anthropogenic aerosols across these scenarios. We also evaluate the model's performance against surface and satellite observations. We build upon and expand on previous studies by incorporating simulations of $PM_{2.5}$, its associated radiative forcing, and health impacts across a comprehensive range of scenarios, including SSPs, ECLIPSE, and the United Nations Environment Programme (UNEP) Integrated Assessment of Air Pollution and Climate Change in Africa pathways, to explore a broad spectrum of future possibilities under plausible conditions. To achieve this, we use a chemical transport model with a more detailed representation of the chemistry involved in the formation of $PM_{2.5}$, compared to Earth System Models which must sacrifice detailed online chemical mechanisms for computational efficiency.

L95: more detailed representation compared to what?
Thank you for your comment. We revised the sentence in the manuscript to the following: To achieve this, we use a chemical transport model with a more detailed representation of the chemistry involved in the formation of $PM_{2.5}$, compared to Earth System Models which must sacrifice detailed online chemical mechanisms for computational efficiency.

L98: "observed and satellite data" might imply the satellite data isn't observed? Just a quick rephrasing needed for clarity.
We have revised this in the manuscript to "surface and satellite observations"
L104: "between 2050 and 2015-2018" – switch these around I think for ease of reading; should be "from past to future".
Thank you for the suggestion. We reworded the sentence in the manuscript to the following: "between 2015-2018 and 2050"

Methods
L108: worth defining WHO, and also for reproducibility a brief explanation of those minor adjustments, in the supplement.
Thank you for the helpful suggestion. We have now defined WHO as the World Health Organization upon first mention. Additionally, we have included the following brief explanation of the minor adjustments made to the WHO African regions in the Supplementary Information: African regional groupings in **Figure 1** are adapted from the World Health Organization (WHO) classifications, with minor adjustments to include all continental countries. Specifically, Sudan, South Sudan, and Somalia were reclassified from the WHO Eastern Mediterranean Region to Eastern Africa, and Libya, Egypt, Tunisia, Morocco, and Western Sahara were reclassified to Northern Africa. These countries are considered part of the Eastern Mediterranean Region in the original WHO classifications but are geographically part of the African continent.

L114: not sure it's worth stating your overall method again here.
To address this, we deleted the following sentence in the manuscript: "We examine the implications of projected mid-century anthropogenic air pollution levels across Africa, along with their associated health and climate impacts."

Figure 1: I think the SSA is referring to the country boundaries being a different colour, but this isn't obvious. It would be best to just have the border round the actual region be distinct, and not the internal borders. Also clarify somewhere that sub-Saharan = West + East + Central + South (if I've understood that right).

Based on your observation that focusing solely on sub-Saharan Africa may undersell the broader value of the study. In response to your later suggestion at L283, we have expanded the study to cover the entire African continent and revised Figure 1 accordingly. The term "sub-Saharan Africa" has been removed where no longer appropriate.

L122: see above on SSP scenario names. It would probably be better to list the scenarios in 3 bullet points, maybe 1 for each source ie SSPs, ECLIPSE, UNEP? Use specific names – perhaps shorthand ones – and use these throughout the paper.

To address this, we revised the sentence in the manuscript to the following: We use both baseline (2015-2018) emissions and projected emissions for the year 2050 from ten available pathways:

- SSP1-1.9, SSP2-4.5, SSP3-7.0, and SSP5-8.5 (hereafter referred to as SSP119, SSP245, SSP370, and SSP585, respectively) (Riahi et al., 2017);
- ECLIPSE version 6b (ECL6) Current Legislation (ECL6 CLE), ECL6 Maximum Feasible Reduction (ECL6 MFR), and ECL6 Sustainable Development Scenario (ECL6 SDS) (IIASA, 2024);
- UNEP Baseline (UNEP BASE), UNEP Short-Lived Climate Pollutants (UNEP SLCP), and UNEP Agenda 2063 (UNEP 2063) scenarios (UNEP, 2022).

To maintain consistency with the figure labels and improve clarity, we refer to the SSP scenarios without the punctuation (e.g., SSP119, SSP245) throughout the manuscript.

L127: what is the resolution of these new emissions? How does this vary across the emissions inventories?

The spatial resolution of the ECLIPSE version 6b emissions is 0.5° latitude by 0.5° longitude. We revised the sentence in the manuscript to: The ECL6 Version 6b covers the years 1990 - 2050 and includes gridded aerosol and reactive gas emissions at 5 and 10-year intervals, provided at a spatial resolution of 0.5° latitude by 0.5° longitude.

All other emissions inventories are also provided at the same 0.5° latitude by 0.5° longitude resolution, and we have mentioned their resolutions in the manuscript.

L146: "[to provide emissions] for 2018 and 2050" – would suggest this, for clarity

Thank you for the suggestion. We have revised the sentence in the manuscript to: We used the three UNEP scenarios: Baseline, SLCP, and Agenda 2063 to provide emissions for 2018 and 2050

L148: "be developed, in contrast" – this comma should be a full stop or semi colon.

Thank you for pointing this out. We have replaced the comma with a full stop.

L150: ", like advances" – would replace "like", maybe "such as"?

Thank you for the suggestion. We have replaced "like" with "such as" to maintain formal tone and clarity.

L160: would use semi-colons to separate this list, for clarity.

Thank you for the suggestion. We have revised the sentence to use semicolons

L164: "SSPs" to "SSP"

We have corrected "SSPs" to "SSP".

L165: do you not need a paragraph on the ECLIPSE emissions? Or is the info in the first paragraph of this section sufficient (I think it could be expanded to explain the motivation of these emissions)? In any case, I would put this info in separate paragraphs, in the order in which you introduce the scenarios at the start of this section, to aid the reader. I think you need

to be clearer on which species are included in each inventory – do they all give the same?

Thank you for this constructive suggestion. In response, we have revised this section to include a dedicated paragraph on the ECLIPSE emissions, providing additional context on the motivation and structure of the scenarios (CLE, SDS, and MFR). We also reorganized the emissions section into clearly separated paragraphs for each scenario family (SSP, ECLIPSE, UNEP), in the same order in which they are introduced earlier in the manuscript, to improve clarity and readability. Additionally, we have clarified the specific species included in each inventory.

L168: Don't need "The" at the start.
"The" has been removed.
L178: would drop the word "recently"; as long as it's the documentation for the model you used, this doesn't matter, and 2018 is debatable as recent in a fast-moving field!
We have removed the word "recently" from this sentence.
L180: can you say roughly which fields are initialised in the model?
To address this, we have added the following in the manuscript: The model is initialized with fields including temperature, precipitation, relative humidity, pressure, cloud water, boundary-layer turbulence, advective and convective mass fluxes.

L180: I think this should be a new section, on "Experiments" – since you have information on the simulations, rather than just the model itself. Readers looking for the experimental setup won't necessarily expect it to be in the model documentation section.
Thank you for this helpful suggestion. We have moved the description of the simulations to a new section titled "Experiments" to clearly distinguish the model description from the experimental setup.
L191: technically 2015 is the first "future" year for the SSPs, right? So maybe drop the bit in brackets.
Thank you for pointing this out. You are correct. 2015 marks the beginning of the scenario period in the SSP framework and is not strictly part of the recent historical emissions record. We have removed the phrase "(i.e. most recent historical)" from the sentence. The updated sentence now reads: "The available emission scenarios have somewhat different baseline years: 2015 for the SSPs, 2016 for ECL6 emissions, and 2018 for UNEP emissions."
L194: this 2% is not just due to the different time period, but also the inventories themselves, right? This strengthens your justification for using different baseline years.
Thank you for pointing that out. We agree that the 2% difference is due to both the different time periods and the inventories themselves. We revised the sentence in the manuscript to read: The difference in simulated $PM_{2.5}$ concentrations between these three base years is 2 % on a continent-wide annual average (**Figure S3**), suggesting limited influence on our analysis from the differing time period and inventories considered
L195: info on this should just be in Section 2.4.
Thank you for your suggestion. We have moved the information about the model evaluation to Section 2.4 as you recommended.
L200: "[and] ammonium". Is the fine-mode defined with an upper level at 2.5um? i.e. giving exactly PM2.5? This is important to clarify.
Yes, the fine-mode is defined with an upper cutoff at 2.5 μm, which corresponds directly to $PM_{2.5}$. We have added the following in the manuscript: The fine-mode is defined with an upper cutoff at 2.5 μm, which corresponds directly to $PM_{2.5}$.
L203: need "-" between "population weighted"
We have added the hyphen between "population" and "weighted".
L204: I would drop the 2015, 2016, 2018 as you have already described this; just refer to something like "respective baselines" or even just "baselines".

We have dropped the 2015, 2016, 2018 and the sentence now reads: We analyzed the differences in population- weighted PM$_{2.5}$ between future emissions and their respective baselines depending on the scenario as described above.

L206: population isn't an output from ISIMIP "simulations"; it's an input – so just clarify that.

We revised the sentence in the manuscript to read: "Annual global population data at a 0.5°resolution, which is an input in the Inter-Sectoral Impact Model Intercomparison Project (ISIMIP2b) simulations (Jones and O'Neill, 2016), was used in our analysis"

L207: minor point but just say 1 degree resolution – you didn't specify lat and lon in the 0.5 deg population above. Thank you for your suggestion. We have now specified latitude and longitude for the population resolution and ensured consistency by using the "latitude by longitude" format throughout the manuscript.

L209: I'm not sure what this sentence means – the model could be ran in 1 degree resolution? I'm not sure how that shows the downscaling works well?

We agree that the original sentence was unclear. To address this, we updated **Figure S1** to show comparisons between modelled and observed surface PM$_{2.5}$ at both the native (2.25° latitude by 2.25° longitude) and downscaled (1° latitude by 1° longitude) resolutions. We revised the text in the manuscript to the following:

The downscaling provides sufficient spatial resolution to capture major spatial variations in both PM$_{2.5}$ concentrations and population distributions. Comparison of modelled and observed surface PM$_{2.5}$ shows that the downscaled 1° latitude by 1° longitude resolution performs slightly better than the native 2.25° latitude by 2.25° longitude model resolution, when averaged over Africa. However, city-level performance varies by location (**Figure S1**).

L212: how did you assign gridcells to countries, when borders will cut across gridcells? Just need to clarify the method used.

Thank you for this useful comment. We have clarified in the text that grid cells were assigned to countries based on whether their center point lies within national borders, using a shapefile of administrative boundaries. We added the following to the manuscript:

Grid cells were assigned to countries based on whether their center point lies within national boundaries, using a shapefile of administrative boundaries from the Natural Earth dataset.

L214: "SSP population" not "SSPs.. ". This approach seems sensible, as you have the population data for SSPs but presumably there aren't the same for the others, but it's worth briefly exploring the sensitivity to this – since this is an inconsistency in the other scenarios. i.e. do you get the same overall results if you just use SSP2 for the SSPs as well?

We have corrected SSPs to SSP. Thank you for this suggestion. To address this, we recalculated population-weighted PM$_{2.5}$ using only SSP2 population data across all scenarios. We found that the differences were negligible (typically <0.1 µg m$^{-3}$), with no changes to the overall trends or scenario rankings. This confirms that our findings are not sensitive to the population dataset used. We added the following to the manuscript:

To assess sensitivity to the population dataset, we repeated the calculations using SSP2 population data for all scenarios. The resulting population-weighted PM$_{2.5}$ values differed by less than 0.1 µg m$^{-3}$ on average, with no change to the relative scenario rankings or spatial patterns. This indicates that the population-weighted PM$_{2.5}$ estimates are not sensitive to the choice of SSP population dataset.

L243: what does level 3 mean? If not relevant, can leave out.

The mention of "level-3" refers to the fact that MYD08_D3 is a gridded, quality-controlled product derived from swath-level (Level-2) observations. However, since this distinction is not directly relevant to our analysis, we have removed the term "level-3" for clarity.

L244: see above for consistency in resolution formatting.

The resolution format has been made consistent.

L251: not sure about using "our" here – maybe clarify the research group? No strong opinion on this though.

We revised the sentence to remove 'our' since we have already referenced our previous studies (Chowdhury et al., 2022, 2024). The revised sentence now reads: In combination with the generated data on ambient PM$_{2.5}$ exposure, we apply the MR-BRT (meta-regression-Bayesian, regularized, trimmed) exposure-response function (Murray et al., 2020; Pozzer et al., 2023), which was also used in previous studies (Chowdhury et al., 2022, 2024) and the most recent iteration of the Global Burden of Disease study (GBD, 2021).

L252: reference for the recent GBD study?

How do you deal with the uncertainty in this function? Do you take draws? What about the low concentration threshold below which there is no impact? There also should be something brief on how this function was made, and its applicability for Africa.

We have referenced the most recent GBD study, which is GBD 2021.

To address this, we added the following in the manuscript: "To account for uncertainty, we drew multiple samples from the posterior distribution of the exposure-response function. We assumed a theoretical minimum risk exposure level (TMREL) ranging from 2.4 µg m$^{-3}$ (Murray et al., 2020). Below this threshold, no further reduction in mortality is assumed. The MR-BRT function was developed using global cohort data, including studies from high-exposure settings, such as Li et al. (2018), Yang et al. (2018), Yin et al. (2017), and Yusuf et al. (2020), enhancing its relevance for regions with high PM$_{2.5}$ levels, such as Africa (Chowdhury et al., 2022). Additionally, we incorporated uncertainty in baseline mortality rates reported by the respective countries. Full methodological details can be found in Chowdhury et al. (2020) https://iopscience.iop.org/article/10.1088/1748-9326/ab8334, and are not repeated here."

L262: this paragraph is a bit harder to follow; I would define the terms first and then expand on them. The use of a 25km2 grid needs more justification – where does this come from? Is it meaningful to downscale the CTM results to this extent, or would it be better to upscale?

We have clarified the definitions of all key variables at the start of the paragraph to improve readability, following the suggested define-then-expand structure. Additionally, the reference to "25 km$^{2}$" was a typo. It should have read "25 km by 25 km horizontal resolution", which we have now corrected in the manuscript. The downscaled resolutions (1° latitude by 1° longitude and 25 km by 25 km) perform similarly to the native resolution, with performance metrics of the same order of magnitude, though slightly better over Africa overall (**Figure S1, Table S2**). However, at the city level, performance varies by location. Beyond 25 km by 25 km resolution is more of a stretch given that concentrations will just be interpolated from rather coarse resolution, and we won't pick up spatial details.

L275: 5km2 is a typo?

Yes, this is a typo. It should have read "25 km by 25 km horizontal resolution". It has been corrected

L277: this use of SSP2 was mentioned above, without this justification; this should be included in that earlier part, with a reference.

Thank you. We have added this justification earlier in the manuscript and cited the UNEP (2022) assessment report, which clearly states the use of SSP2 population projections.

We revised the previous sentence in the manuscript as follows:

SSP2 population data was used for UNEP and ECLIPSE cases, as SSP2 aligns closely with UN population projections (UNEP, 2022), while the respective SSP population projections were matched for the other SSP scenarios.

Results

L283: why "sub-Saharan"? This is the same as all the non-north Africa regions I guess? I think Figure 1 could be improved to be more clear on this. There is a lot of variation in North Africa too, so I think you undersell yourself by focusing on a subset of the continent here.

Thank you for this important comment. We agree that limiting the focus to sub-Saharan Africa could underrepresent the full spatial diversity of air pollution and health impacts across the continent, including in North Africa. In response, we have revised the study to cover all of Africa (i.e., sub-Saharan Africa plus Northern Africa to make up the entire African continent). **Figure 1, Figure 2, Figure 6, Figure S4, and Table S2** were the only figures and table that originally showed data for sub-Saharan Africa. These have been updated to reflect data for the entire African continent. Corresponding changes have also been made throughout the text, and the term "sub-Saharan Africa" has been removed where no longer appropriate. The remaining figures already represented the whole African continent. To improve clarity, the "SAF" label has been removed from **Figure 8, Figure S15, and Figure S16.**

Figure 2: Is "ECL6" a duplication? It's not distinct from others, and there are 11 scenarios in the legend but you mention using 10 above. SAF = Sub-Saharan Africa? That needs defining somewhere; if you are going to use that then you can include it in figure 1. But maybe it's worth just writing it out in full like the other regions. Note in the caption that the y scales vary between regions. It's interesting that CEDSv21 is lower than 17, right? Also that formatting "CEDSv21" is different from the main text.

Thank you for pointing this out. ECL6 is not a duplication. It refers to the historical emissions scenario and has now been renamed "ECL6_Hist" to indicate that it is the historical ECL6. The future scenario, **ECL6_CLE**, continues from 2016 onward.

Following your suggestion in L283 above, we have updated the study to focus on the entire Africa. "sub-Saharan Africa" plot in the future as been updated to reflect data for all of Africa. We have noted that the $y$-axis scales vary between regions.

Yes. Studies have suggested that CEDS version released in 2016 (and also the version released in 2017) overestimated carbonaceous aerosols in Africa (IPCC AR6 WG1 Ch.6, section 6.2.1). CEDS21 made changes to CEDS aligning more with GAINS ECL6 emissions as discussed in Lund et al. (2023). We mainly discuss Asia in that paper, but we suspect this could have affected African emissions.

CEDS formatting has been made consistent. We now use "CEDS21" throughout the manuscript.

L294: SSP245 and SSP585? Also check formatting – it was SSP2-4.5 earlier. Just be consistent.

The SSP formatting has been made consistent. We now use the shorthand form "SSP245, SSP585" throughout the manuscript.

L300: I don't think Wells et al., 2024 focused on SSP585 – they mention $SO_2$ emissions in SSP119 being driven by industry, which is relevant to your next sentence, but this reference to BC in SSP585 is worth checking. That next sentence isn't quite right either – $SO_2$ emissions increase to mid-century in SAF in SSP119 and SSP585, before decreasing; you might be saying this already but worth clarifying for sure.

Thank you for this helpful comment. You are correct that Wells et al. (2024) primarily discuss $SO_2$ emissions in the context of SSP119 and not BC emissions in SSP585. We have now reference BC emissions in SSP585 using Turnock et al. (2020) paper which looked at the emissions in different regions (including Africa) under the SSPs. The next sentence has been condensed into the $SO_2$ emissions paragraph, and Wells et al. (2024) is now cited where appropriate. We have also clarified the description of $SO_2$ emissions, emphasizing their increase to mid-century and subsequent decline in SSP585 and SSP119. We added the following to the manuscript:

After mid-century, BC emissions in Africa are projected to decline under the SSP245 and SSP585 scenarios, driven by a reduction in residential fossil fuel use (Turnock et al., 2020). For $SO_2$ emissions, the overall pattern in Africa shows projected increases under all scenarios by 2050 except for the ECLIPSE scenarios, with declines after mid-century under SSP119, SSP245, and SSP585.

Revised sentence where Wells et al. (2024) is cited:
"For instance, Central Africa and Southern Africa $SO_2$ emissions are projected to increase by 303% and 47% by 2050 under the SSP119 scenario, driven by reliance on coal during the transition period (Wells et al., 2024)."
L308: would simplify by saying "the ECLIPSE scenarios"; generally, this section could do with a bit of tightening up just to simplify and aid the reader. Currently it's a bit overwhelming and hard to take in all the detail; the reader would take away more if there was less info. Figure out the key trends you want to highlight, and consider dropping some others.
It has been simplified to "the ECLIPSE scenarios". Section 3.1 has been substantially shortened and every attempt has been made to ensure salient points are coming through.
L310: to 2050 right?
Yes, to 2050. It has been clarified
L318: needs a reference here; also worth noting this dynamic (reducing warming SLCPs but not SO2) earlier when you introduce the scenarios.
We have added a citation at L318 to clarify the selective mitigation approach of the SLCP scenario, referencing UNEP (2022). We have also clarified this dynamic earlier in the text (Emissions section) where we introduce the SLCP scenario: We revised the earlier text in the manuscript to the following:
In contrast, the SLCP scenario calls for the implementation of mitigation measures that specifically target warming SLCPs, such as BC (UNEP, 2022). Importantly, this scenario does not address $SO_2$ emissions, which have a cooling effect (UNEP, 2022).
L337: "the SSP370" – needs rephrasing. Again, lots of information in this bit which the reader may struggle to absorb (speaking from experience!).
Thank you for pointing this out. We rephrased the sentence to read:
By 2050, BC and $SO_2$ emissions in Northern Africa are projected to increase substantially under SSP370. Section 3.1 has been substantially shortened and every attempt has been made to ensure salient points are coming through.

L361: could do with explaining exactly why it's important for accurate sectoral information (i.e. IAM consistency?).
Thank you for your comment. We have clarified the importance of accurate sectoral data, explicitly linking it to IAM consistency. We added the following in the manuscript: This is essential for ensuring consistency across integrated assessment models (IAMs) and for better constraining scenario baselines by grounding models in the most current and reliable emissions data and sectoral activity patterns.

Figures 3+4: these are nice way of showing this, but these need some work. The figures are currently squashed making them harder to read. Maybe you could have one comparative stacked bar chart for each region instead of the 3 pie charts? Like the supplement. But maybe that won't work. List the "increasing scenarios" in the caption, not just the main text.
The increasing scenarios are now listed in the captions: The revised captions now read:
Thank you for your comment. Figures 3 and 4 have been updated to improve clarity and resolution.
**Figure 3:** Sub-regional sectoral contributions to BC emissions (kilotons per year) in 2020 and 2050 under the increasing scenarios SSP370, ECL6 CLE, and UNEP Baseline.

**Figure 4:** Sub-regional sectoral contributions to $SO_2$ emissions (kilotons per year) in 2020 and 2050 under the increasing scenarios SSP370, ECL6 CLE, and UNEP Baseline.

L376: "the SSP585"
We have removed the 'the' before 'SSP585'.

Section 3.2 has some really nice material, and is clearer than the previous section. E.g. the last paragraph, L367-9, L388-90 are clear; it's great when you can put these differences in their broader context. This still needs some work though to condense out the key narratives you want to get across. In general you should briefly say what the supplementary figures show as well as the conclusions drawn from them. You slip into discussing "the SSPs" when you are only looking at SSP370 in this bit, right? Need to be careful on that.
We have revised the manuscript to briefly say what the supplementary figures show and to clearly state the conclusions drawn from them. We have also ensured that the discussion in this section is focused on SSP370 where applicable and generalized only when appropriate. Section 3.2 has been substantially shortened and every attempt has been made to ensure salient points are coming through.

L437: this sounds like you're referring to prior analysis; need to be clear these are your results.
We thank you for this helpful suggestion. We revised the sentence to the following: We validated the OsloCTM3 model against $PM_{2.5}$ surface observations for 2019 and MODIS Aqua AOD data across Africa, finding a broad range of performance across regions.
L439: define MAE, RMSE. Maybe make the point here that there is some disagreement between the observations? And set your results in that context.
We appreciate your suggestion to define MAE and RMSE. These terms are already defined in **Section 2.5** of the methodology. We also agree with your point regarding the disagreement between the observations. We added the following to the manuscript: There is some disagreement between the observational datasets themselves, which should be considered when interpreting model performance.

Figure 5: this needs tidying up too; the maps are small and the plot could be better formatted. Maybe use a discrete scale to make it easier to read roughly what each value is.
Again, you should introduce the figures in the text with roughly what they show, so the reader doesn't have to jump to the figure straight away to get the picture.
Thank you for pointing this out. We have updated **Figure 5** to improve readability and layout. We have enlarged the map panels for better visual clarity, reformatted the figure layout for consistency and better balance, and applied a discrete color scale to make it easier to read roughly what each value is. We have also introduced the figures in the text with roughly what they show, so the reader doesn't have to jump to the figure straight away to get the picture.
Figure 6: you should use the same regional colour scheme as in the maps for these lines; I think you've used close colours but they're not exactly the same. Maybe it would be worth adding a second panel showing the percentage changes, as these are not the same as the magnitude ones and also tell an interesting story?
Thank you for pointing this out. We revised the figures to use the same color for the regions. Now regional colors in the map, in **Figure 6**, and in **Figure 7** have been made consistent. We updated **Figure 1** and **Figure 6** using the same color codes used in **Figure 7**. In addition, we have added a second panel to the **Figure 6** showing the percentage changes in $PM_{2.5}$ concentrations, which provides further insight beyond the absolute values.
L468: "exposure [increase]"?
We have revised the sentence to clarify that it refers to $PM_{2.5}$ exposure increase. We revised the sentence to: Western, Eastern, and Northern Africa are projected to experience the most substantial $PM_{2.5}$ exposure increase under the UNEP Baseline, SSP370, and ECL6 CLE

scenarios due to increased emissions from residential, industrial, transportation, and energy sectors.

L483: you need to show that this is the case – see suggestion for figure 6 above.

Thank you for the suggestion. **Figure 6** has been updated to show relative (percentage) changes in $PM_{2.5}$ concentrations, and this has now been referenced appropriately in the text.

L486: this is nice – relating your results to the underlying drivers.

Thank you for the positive feedback. We are glad that this connection between projected health benefits and underlying emission sources comes across clearly.

L492: you use "MFR" here instead of "ECL6 MFR" – it's fine to have shorthand (and maybe this would be a good thing to use throughout) but need to be consistent. If you use shorthand names, then define them when you introduce the scenarios – see the suggested in the intro about bullet points for these scenarios.

Thank you for pointing this out. We have corrected "MFR" to "ECL6 MFR" for consistency and have ensured that all scenario shorthand names are defined when the scenarios are first introduced in the emissions section.

L494: don't use PM as shorthand for PM2.5; you need to write it out in full if this is the metric we are looking at.

Thank you for the comment. We have replaced "PM" with "$PM_{2.5}$"

L511: "s partially due" -> primarily due, right?

Thank you for the suggestion. We have revised the phrase to: There are large increases in almost all cases primarily due to the projected population increase (**Figure S13).**

L512: "Western African" - Western Africa's? also, need to be clear this is SSP370 (right?), as there is wide variation in the figure you refer to.

Thank you for pointing this out. We have clarified that these projections refer to the SSP370 scenario and revised "Western African" to "Western Africa's". We revised the sentence to the following: Africa's population is expected to double by 2050 under SSP370, growing from 1.1 billion to 2.3 billion, with Western Africa's population reaching 0.9 billion, followed by Eastern Africa with 0.7 billion, Southern Africa with 0.26 billion, Central Africa with 0.24 billion, and Northern Africa with 0.22 billion, as illustrated in **Figure S11**.

Section 3.4: this section was a bit of an anticlimax – this is one of your main results, and it's only 2 paragraphs! Figure S13 is really interesting, showcasing the different drivers of these changes – and that these are fairly consistent across regions, with some important scenario and regional differences. Figure 7 is great (though the colours don't seem quite the same as figure 6, and the order is different, which should just be fixed), but you don't refer to it in the text at all! Though you do quote data from it.

Thank you for pointing this out. We have updated **Figure 1** and **Figure 6** using the same color codes in **Figure 7**, and we have also aligned the ordering of regions and scenarios across **Figure 6** and **Figure 7**. Additionally, **Figure 7** is now explicitly referenced in the main text.

We have expanded the health section to further discuss the key drivers of projected excess deaths as shown in Figure S13, highlighting consistent patterns across regions as well as important scenario- and region-specific differences. We added the following paragraphs in the manuscript:

Aging, a key aspect of the demographic transition, contributes to increased excess deaths due to age-related diseases, even with modest population growth. In contrast, overall population growth amplifies excess deaths, particularly in regions with high fertility rates and younger populations such as Western Africa and Eastern Africa. In Northern Africa, while population growth is projected to be slower, the aging population resulting from demographic shifts is projected to lead to a substantial number of excess deaths, reaching 300,000 under SSP119 and SSP585, and exceeding 200,000 in other scenarios (**Figure S13**). Southern Africa is projected to experience the lowest number of excess deaths attributable to aging,

except under SSP119 and SSP585, where demographic transition-related excess deaths increase to about 20,000.

As illustrated in **Figure S13**, Southern Africa is projected to benefit significantly from the pollution transition, with reductions in pollution-related deaths approaching 10,000 under scenarios such as SSP119, SSP585, ECL6 SDS, and ECL6 MFR, primarily driven by improvements in industrial emission controls. In other regions, the reductions are more modest, reflecting varying levels of emissions control and air quality improvements. Most regions are projected to benefit from the epidemiological transition, characterized by a shift from infectious diseases to non-communicable diseases (NCDs). Excess deaths associated with this transition are projected to decline across most regions and scenarios, except Central Africa under SSP119 and SSP585, where increases are projected. The most substantial reductions are projected under SSP119 and SSP585, with Western Africa, Southern Africa, and Northern Africa each expected to experience declines of up to about 100,000 excess deaths.

L531: I don't think this conclusion about baseline concentrations follows, unless you make an argument based on the functional form of the response function? The population increase would drive increases regardless of the baseline, all else being equal.

Thank you for pointing this out. Yes, you have a valid point. As you note, population increases would drive increases regardless of baseline $PM_{2.5}$ levels. Risk in the response function flattens at higher $PM_{2.5}$ concentrations. We have removed that conclusion: "This further highlight that there are already high $PM_{2.5}$ concentrations due to fires and natural dust (Pai et al., 2022).**"**

Figure 7: are these error bars from the response function?

Thank you for your comment. Yes, the error bars in **Figure 7** represent the 95% confidence intervals derived from the uncertainty in the exposure-response function (MR-BRT). We have revised the caption to the following: **Figure 7:** Excess deaths due to $PM_{2.5}$ calculated for baseline and 2050 emissions for each region and scenario. The black bars represent error bars, showing the 95% confidence interval in the excess death estimates for each region and scenario, reflecting uncertainty propagated from the exposure-response function (MR-BRT).

"3.5 Anthropogenic aerosol-inducing radiative forcing" – aerosol-induced?

Thank you for pointing this out. We have corrected the section title to "Anthropogenic aerosol-induced radiative forcing"

L542: "s between 2050 and 2015, 2016 or 2018," – again, just say baseline, and go from present to future.

Thank you for the suggestion. We have revised the sentence to use "baseline" and went from present to future. We revised the sentence to the following: In **Figure 8**, we show the net radiative forcing due to changes in anthropogenic aerosol emissions between the baseline and 2050, depending on the emission scenario.

L544: should note again here that this is only the 1$^{st}$ cloud effect, not the 2$^{nd}$ (lifetime) one. also just name again the method/model used.

Thank you for pointing this out. We have clarified in the manuscript that the aerosol–cloud interactions refer only to the first indirect effect (cloud albedo effect), not the second (cloud lifetime effect). We have also named again the method/model used. We revised the sentence to the following: This includes the radiative forcing due to aerosol-radiation interactions (RFari, shown in **Figure S15**), estimated using offline radiative transfer calculations with a multi-stream model based on the discrete ordinate method (Myhre et al., 2013; Stamnes et al., 1988) and from aerosol-cloud interactions (RFaci, shown in **Figure S16**), estimated using a

parameterization based on Quaas et al. (2006). The aerosol-cloud interaction estimate includes only the first indirect effect (cloud albedo effect), not the second (cloud lifetime effect).

L551: clarify regional-mean is SAF.

Thank you for your suggestion. We have clarified in the text that the mean values refer to the entire African continent rather than a specific region, to avoid ambiguity. The sentence now reads: A net positive aerosol forcing in 2050 relative to the baseline is projected in all but two scenarios, with continent-wide mean values ranging from 0.03 W m$^{-2}$ in SSP119 to 0.27 W m$^{-2}$ in SSP585.

L559: "emissions of changes" -> "changes of emissions"?

Thank you for pointing this out. We agree and have revised the sentence for clarity.

L552: before you go into detail here, just state the overall result: you generally find a small warming effect due to reduced concentrations, with cooling from the increase in 2 scenarios.

Thank you for your suggestion. We added the following to the manuscript: Overall, we find a small warming effect due to reduced aerosol concentrations, with cooling in the two scenarios where emissions increase.

L584: "CEDS is not updated to 2022" – not sure when this sentence was written but CEDS is available to 2022 now; maybe just note a general delay in inventories?

Thank you for the clarification. We have revised the sentence to reflect the current status of CEDS and to emphasize the broader issue of delayed updates and lack of harmonization across inventories. The updated text now reads: The recent emission inventories differ and often lack harmonization with scenario data; although CEDS has now been extended to 2022, there is generally a delay in updating inventories and aligning baselines, highlighting the need for improved data to enhance inventory construction and reduce uncertainties.

L636: "significant weakening of aerosol radiative forcing by 2100 across all scenarios" – is this true spatially, e.g. in Africa? Or just globally?

Thank you for your comment. You are correct to point out the potential spatial variation in aerosol radiative forcing. The statement regarding the significant weakening of aerosol radiative forcing by 2100 applies primarily on a global scale. However, as noted by Lund et al. (2019), there are substantial regional differences, particularly in regions such as South Asia and Africa, where the direction and magnitude of aerosol forcing may vary depending on the scenario. We have added the following to the manuscript: "While present-day and future radiative forcing estimates vary across studies, a consistent global finding is the significant weakening of aerosol radiative forcing by 2100 across all scenarios (Lund et al., 2019). However, Lund et al. (2019) also highlighted substantial regional differences, particularly in South Asia and Africa, where the magnitude and even the direction of forcing change vary depending on the scenario."

L640: you should bring in CMIP6 studies on aerosol forcing e.g. https://acp.copernicus.org/articles/21/853/2021/ - to display the inter-model variability.

Thank you for your suggestion. We have incorporated a discussion of the inter-model variability in aerosol forcing from CMIP studies, as requested.

We added the following to the manuscript: This inter-model variability is further highlighted by results from the Coupled Model Intercomparison Project Phase 6 (CMIP6), where Thornhill et al. (2021) show large differences in aerosol effective radiative forcing across models, reflecting substantial uncertainty in how aerosol-climate interactions are represented.

L649: "found that BC absorbs short-wave radiation," – careful with your phrasing when it comes to describing other studies; they didn't show this effect (ie for the first time); they look at the implications of this effect.

It's worth noting in the discussion that your method will miss climate driven PM2.5 changes, though these shouldn't be very large. Also you should note that while you explore uncertainty in emissions, your use of one model means the uncertainty in model response – concentrations and forcing - is not explored.

Thank you for the suggestion. We have revised the sentence to clarify that Myhre et al. (2017) examined the implications of black carbon absorbing short-wave radiation, rather than identifying the effect itself. The updated sentence now reads: "For instance, Myhre et al. (2017) examined the implications of BC absorbing short-wave radiation, highlighting its potential to warm the atmosphere and cause contrasting effects on circulation patterns and regional precipitation."

We acknowledge in the discussion that our approach does not capture climate-driven changes in $PM_{2.5}$ and that using a single model limits exploration of uncertainty in model response, though emission uncertainty is addressed. We added the following to the manuscript: "Our approach does not capture climate-driven changes in $PM_{2.5}$, which are expected to be smaller than emission-driven changes (Westervelt et al., 2016). Additionally, while we assess uncertainty from emissions, using a single model means we do not explore uncertainties in model response, such as differences in simulated concentrations and forcing."

L709: this is the first time you've mentioned aging as a cause of this increase – how important is this cf the general population increase? If it's important, then your earlier presentation of results as being driven by population change should be clarified to note this is not just a numerical population increase, but also the aging effect.

Thank you for this insightful comment. Yes, although the primary driver is the numerical population growth, aging, as a key aspect of demographic transition, also plays an important role in driving projected excess deaths. To address this, we added a paragraph to discuss the overall effect of aging vs population growth. We added the following to the health section:

Aging, a key aspect of the demographic transition, contributes to increased excess deaths due to age-related diseases, even with modest population growth. In contrast, overall population growth amplifies excess deaths, particularly in regions with high fertility rates and younger populations such as Western Africa and Eastern Africa. In Northern Africa, while population growth is projected to be slower, the aging population resulting from demographic shifts is projected to lead to a substantial number of excess deaths, reaching 300,000 under SSP119 and SSP585, and exceeding 200,000 in other scenarios (**Figure S13**). Southern Africa is projected to experience the lowest number of excess deaths attributable to aging, except under SSP119 and SSP585, where demographic transition-related excess deaths increase to about 20,000.

L714: "projected to reduce excess deaths" – should be clear this is just the tendency of these effects, not the overall effect (which is driven by population)
We have revised the phrasing to clarify that the reduction in excess deaths due to epidemiological and pollution transitions represents a tendency, not the final overall effect. The final outcome is primarily driven by population growth. We revised the sentence to: Epidemiological and pollution transitions are projected to tend to reduce excess deaths in all regions, particularly in Western, Southern, and Northern Africa, with notable reductions in Southern Africa

Referee: 2
**General comments:**

This manuscript presents a valuable assessment of future aerosol emission projections for sub-Saharan Africa, highlighting the considerable diversity among scenarios and their

implications for air pollution, health, and climate in the future. Overall, the manuscript is well-structured and presents interesting and much-needed results that contribute to understanding future emissions in Africa and their potential health impacts.

However, there are several areas that could be improved. The introduction feels somewhat disjointed, and although one of the main focuses of the article is the impact of air pollution on health, this topic is only briefly addressed. My primary recommendation is to expand on this aspect within the introduction and improve the overall narrative flow.

In addition, certain methodological details could benefit from further clarification. For example, in Section 2.2, where all scenarios are defined, it is at times difficult to follow which and how many scenarios are actually used. Moreover, the rationale behind the selection of scenarios should be better justified.

Some figures also lack sufficient quality—Figures 3 and 4 in particular would benefit from improved resolution and clarity.

Finally, although the manuscript acknowledges uncertainties in emission inventories, providing more detail on how these uncertainties might influence the main conclusions would offer a clearer perspective on the robustness of the findings.

Addressing these points would significantly enhance the clarity and impact of the manuscript. These comments, along with minor technical corrections, are detailed in the specific and technical comments below. Overall, this study makes a strong contribution to the field and, with some adjustments, will serve as a valuable resource for future research and policy discussions.

We sincerely thank the reviewer for their thoughtful and constructive feedback. We appreciate the recognition of our study's contribution to understanding future aerosol emissions and their implications for air pollution, health, and climate in sub-Saharan Africa. We have revised the introduction to improve its coherence and flow, with a clearer structure that better frames the study objectives. We have also expanded the discussion of air pollution and health impacts to emphasize their importance and relevance to the overall narrative. Section 2.2 has been revised for clarity. We now clearly state the number of scenarios used and have improved the explanation of each. We have also added justification for the selection of scenarios, emphasizing their relevance to Africa-focused development pathways and policy frameworks. Figures 3 and 4 have been updated with higher resolution and improved labeling to enhance clarity and readability. We have expanded our discussion on uncertainties related to emission inventories and how these might influence our results. This includes acknowledgment of uncertainties in model responses and emissions data, and their implications for the robustness of our conclusions. We appreciate your feedback and believe the revisions have strengthened the manuscript.

**Specific comments:**

**Introduction:**

As mentioned previously, the introduction appears somewhat brief and simplistic. It could benefit from improved coherence and a more fluid narrative. Moreover, it lacks sufficient information on the health impacts of air pollution, which is one of the central pillars of this study. Strengthening the connection between changes in air pollution and their health impacts would enhance the cohesion and relevance of the introduction. We thank the reviewer for the comment. To address this, we added: 'Ischemic heart disease (IHD) and stroke have been identified as major contributors to the health burden linked to air pollution (Fang et al., 2025)' to the previous paragraph and also expand by adding a new paragraph as follows:

The health impacts of air pollution become more severe with age because aging

reduces physiological resilience and increases susceptibility to chronic, non-communicable diseases (Pope, 2007; Shumake et al., 2013). Older adults, particularly those aged 60 and above, are especially vulnerable to air pollution (Yin et al., 2021). This age group is growing rapidly, especially in low-income and middle-income countries (LMICs), where pollution levels are typically higher than in wealthier nations (UN, 2017). In addition to older adults, young children, particularly those under five, are also highly vulnerable (Murray et al., 2020). Africa is projected to account for at least 40 % of the global population of children under the age of five by 2050 (KC and Lutz, 2017). At present, about 30 percent of $PM_{2.5}$-related deaths in Africa occur among children under five, and this rate has been increasing over the past decade (Murray et al., 2020).

We have also linked the first paragraph in the introduction which talks about aerosols and climate to the second paragraph which talks about aerosols and health to improve the narrative follow. The layout of the introduction now flows as follows: General/global Aerosol emissions → climate impacts -> health impacts (with greater detail as suggested) → Africa-specific emissions → previous studies on African aerosol → paragraph describing our novel addition to the literature → overall paper layout. We feel this is now a logical layout and flow.

Additionally, the manuscript could better clarify why this study is necessary in comparison to previous research. What are the specific knowledge gaps this study aims to address?

We thank the reviewer for the comment. The manuscript now clarifies that previous studies focusing on Africa such as Shindell et al. (2022) and Wells et al. (2024), have primarily relied on SSP scenarios, limiting the exploration of plausible futures. Our study addresses this gap by using a broader set of scenarios, including ECLIPSE and UNEP pathways, offering a more comprehensive assessment of future climate and health impacts in Africa.

**2. Methods**

**2.1 Study Domain:**

There is insufficient justification for the chosen study domain. For instance:

- **Justification for regional division:** Why were these five specific subregions selected, and how do they influence the analysis?

  To address this, we added the following to the manuscript:

  These divisions are commonly used in regional analyses for ease of interpretation and to reflect the continent's diverse environmental, health, economic, and demographic characteristics (Djeufack Dongmo et al., 2023; Fenta et al., 2020; Safo-Adu et al., 2023). Each region differs in population density and industrialization, which in turn influence air pollution levels, $PM_{2.5}$ exposure, and public health outcomes (Djeufack Dongmo et al., 2023; Safo-Adu et al., 2023).

- **WHO region adjustments:** A brief explanation of why modifications to WHO regions were made and what those adjustments entailed would be helpful.

  Thank you for your suggestion. We revised and added the following brief explanation in the manuscript: The regional boundaries were adapted from the World Health Organization (WHO) Administrative regions for Africa (WHO, 2024), with minor adjustments to include all continental countries and ensure complete geographic coverage of the African continent. For example, several North and East African countries that are geographically part of Africa but classified in the WHO Eastern Mediterranean Region were reassigned to their respective continental sub-regions to better reflect geographic boundaries (see Supplement for details).

- **Definition of boundaries:** How were the boundaries for each subregion determined—based on geographic coordinates, political borders, or climatic characteristics?

  The subregional boundaries were based on the WHO administrative regions, with adjustments (described in detail in the Supplement) to better align with Africa's geographic boundaries. We added the following to the manuscript: The regional boundaries were adapted from the World Health Organization (WHO) Administrative regions for Africa (WHO, 2024), with minor adjustments to include all continental countries and ensure complete geographic coverage of the African continent.

- **References and support:** Including additional references that support the chosen classification or that have used similar regional divisions in previous studies would strengthen this section.

  To address this, we have added references to previous studies that used similar regional groupings to support and strengthen this sedction. References added are: (Djeufack Dongmo et al., 2023; Fenta et al., 2020; Safo-Adu et al., 2023).

**2.2 Emissions:**

This section contains a lot of information on the scenarios and their characteristics. Including a summary table—perhaps in the supplementary material—would help clarify which scenarios are considered. At present, it can be difficult to grasp how many scenarios are used, what they are, and how they differ from each other.

Thank you for your suggestion. We added 'Table 1: Summary of emission scenarios and their characteristics' in the supplement which tells what scenarios we are using and how they differ from each other. We also added '' See Supplementary Table S1 for a summary of the emission scenarios and their characteristics." in the manuscript.

Several scenario frameworks are presented (SSP, ECLIPSE, UNEP), but it is not clearly explained why these particular scenarios were chosen. Why these ten, and not others? A brief justification for the selection of these specific scenarios over other potential alternatives would be valuable.

Thank you for your comment. To address this, we added the following to the methodology to the manuscript: These ten scenarios were selected to span a broad range of plausible futures, capturing variation in socioeconomic development trajectories (SSPs), explicit assumptions about air pollution policy ambition and implementation (ECLIPSE), and Africa-specific development priorities (UNEP). This diverse set enables a comprehensive assessment of the potential climate and health impacts of anthropogenic aerosol emissions across Africa.

Consistency in reference years: The period 2015–2018 is mentioned as the baseline, yet 2016 is used for ECLIPSE and 2018 for UNEP. What accounts for this variation? Clarification would improve the consistency of the methodology.

Thank you for your comment. This has been clarified in the Experiments section (Sect 2.4) as follows: The available emission scenarios have somewhat different baseline years: 2015 for the SSPs, 2016 for ECL6 emissions, and 2018 for UNEP emissions. The difference in simulated $PM_{2.5}$ concentrations between these three base years is 2 % on a continent-wide annual average (**Figure S3**), suggesting limited influence on our analysis from the differing

time period and inventories considered.

**2.3 OsloCTM3**

**Model Simulations and Resolution:** The OsloCTM3 model operates at a horizontal resolution of 2.25° × 2.25°. Is this resolution not rather coarse for capturing spatial details at the regional level? A brief explanation of the advantages or limitations of the OsloCTM3 compared to other chemical transport models would be helpful.

Thank you for the comment. To address this, we added the following in the manuscript: Although this horizontal resolution is relatively coarse, the OsloCTM3 model is well-suited for long-term, large-scale simulations of atmospheric composition. It includes detailed aerosol chemistry and has been extensively applied in both global and regional studies (Aamaas et al., 2017; Lund et al., 2017, 2019; Søvde et al., 2012).

**Explanation of resolution and rescaling:** The manuscript mentions that the model runs at 2.25° × 2.25° resolution, but the data are then rescaled to 1° × 1°. It would be useful to briefly explain why this rescaling is performed and what impact it may have on the accuracy of the results.

Thank you for the comment. To address this, we have revised the manuscript to include the following explanation:

The downscaling provides sufficient spatial resolution to capture major spatial variations in both $PM_{2.5}$ concentrations and population distributions. Comparison of modelled and observed surface $PM_{2.5}$ shows that the downscaled 1° latitude by 1° longitude resolution performs slightly better than the native 2.25° latitude by 2.25° longitude model resolution, when averaged over Africa. However, city-level performance varies by location (**Figure S1**).

**2.4 Observations**

**Justification for the selection of observational data:** The manuscript states that PM2.5 observations are obtained from U.S. embassies in Africa. Is there a specific reason for selecting only these data instead of including other monitoring networks? Clarifying this point would help assess whether this selection introduces any limitations regarding the spatial representativeness of the observational dataset.

Thank you for the comment. We now clarify in the manuscript that U.S. embassy monitors were chosen because they are high-quality reference instruments located in major African cities, providing reasonable spatial coverage for our study. These data are publicly available, and often serve as a standard for local monitoring networks. While some sparse networks in single cities may exist, there is not another unified continent-wide dataset for air quality, and city-scale networks can either be not publicly shared or using less reliable and accurate technology. We added the following to the manuscript: The United States embassy monitors are high-quality reference instruments that provide publicly available, quality-assured $PM_{2.5}$ data. Located in major African cities, they offer reasonable spatial coverage of the most populated and emission-relevant regions in Africa and are often used as a benchmark for calibrating local monitoring networks.

**2.5 Health Impact Assessment**

**Spatial resolution:** Data are interpolated to a 25 × 25 km resolution, but final population estimates are provided at 5 × 5 km resolution. Could this discrepancy introduce biases in the estimates of excess mortality?

Thank you for your comment. The reference to a 5 km by 5 km resolution was a typographical error; all population estimates are provided at a 25 km by 25 km resolution, consistent with the

interpolation resolution. We have corrected this in the manuscript to avoid confusion.

**Results**

Figures 7, S9, and S10 are not cited in the text. Please, add them.

Thank you for your comment. We have now cited Figures 7, S9, and S10 in the main text.

As mentioned earlier, figure clarity should be improved, particularly for Figures 3 and 4. Additionally, in the health-related section, the manuscript does not discuss the relative effect of aging versus overall population growth, an important consideration for interpreting the results.

Figures 3 and 4 clarity and resolution has been improved.

The manuscript now discusses the relative effect of aging versus overall population growth in the health-related section. We added the following to the manuscript:

Aging, a key aspect of the demographic transition, contributes to increased excess deaths due to age-related diseases, even with modest population growth. In contrast, overall population growth will amplify excess deaths, particularly in regions with high fertility rates and younger populations such as Western Africa and Eastern Africa. In Northern Africa, while population growth is projected to be slower, the aging population resulting from demographic shifts is projected to lead to a substantial number of excess deaths, reaching 300,000 under SSP119 and SSP585, and exceeding 200,000 in other scenarios (**Figure S13**). Southern Africa is projected to experience the lowest number of excess deaths attributable to aging, except under SSP119 and SSP585, where demographic transition-related excess deaths increase to about 20,000.

As illustrated in **Figure S13**, Southern Africa is projected to benefit significantly from the pollution transition, with reductions in pollution-related deaths approaching 10,000 under scenarios such as SSP119, SSP585, ECL6 SDS, and ECL6 MFR, primarily driven by improvements in industrial emission controls. In other regions, the reductions are more modest, reflecting varying levels of emissions control and air quality improvements. Most regions are projected to benefit from the epidemiological transition, characterized by a shift from infectious diseases to non-communicable diseases (NCDs). Excess deaths associated with this transition are projected to decline across most regions and scenarios, except Central Africa under SSP119 and SSP585, where increases are projected. The most substantial reductions are projected under SSP119 and SSP585, with Western Africa, Southern Africa, and Northern Africa each expected to experience declines of up to about 100,000 excess deaths.

**MINOR COMMENTS:**

**Terminology consistency:**

- In some parts of the manuscript, the term *"future emissions"* is used, while in others *"2050 emissions"* appears. For greater clarity, it would be helpful to adopt consistent terminology throughout the manuscript.

  Thank you for your comment. We have clarified in the manuscript that "future" refers specifically to the year 2050 by explicitly defining it as "future (2050)" at first mention. We have also reviewed the manuscript to ensure consistent terminology is used throughout.

- Similarly, there is inconsistency in referring to the baseline emissions as *"baseline emissions"* versus *"historical emissions"*. Please ensure consistent use of these terms

across the text.

Thank you for your comment. In the revised manuscript, we have ensured consistent terminology. We now use **"baseline emissions"** to refer specifically to emissions in the defined baseline period, while **"historical emissions"** refers to emissions prior to the baseline period (e.g., from datasets such as CEDS17, CEDS21, and ECLIPSE before 2016). This distinction is now clearly stated in the text.

- **Cross-referencing between sections:** In the section where the model validation against observations in Africa is discussed, it would be helpful to include a cross-reference to Section 2.4 to reinforce the structure and cohesion of the manuscript.

Thank you for the suggestion. We have added a cross-reference to Section 2.4 which is now 'Section 2.5' in the manuscript, the section where the model validation against observations in Africa is discussed. We added the cross reference in the manuscript

as follows: We validated the OsloCTM3 model against $PM_{2.5}$ surface observations for 2019 and MODIS Aqua AOD data across Africa (see Sect. 2.5), finding a broad range of performance across regions.

**Line-specific suggestions:**

- **Line 249:** *"we apply"* → *"we applied"* (to maintain past tense when describing the methods used).

Thank you for catching this. We have revised the sentence to maintain consistent past tense. The sentence now reads: In combination with the generated data on ambient $PM_{2.5}$ exposure, we applied the MR-BRT (meta-regression-Bayesian, regularized, trimmed) exposure-response function (Murray et al., 2020; Pozzer et al., 2023), which was also used in previous studies (Chowdhury et al., 2022, 2024) and the most recent iteration of the Global Burden of Disease study (GBD, 2021).

**Line 252:** *"Using the cause-specific exposure-response function, we estimate excess deaths..."* → Consider revising to *"we estimated excess deaths..."* to align with the use of past tense in the rest of the methodology section.

Thank you. We revised the sentence to: Using the cause-specific exposure-response function, we estimated excess deaths from ischemic heart disease (IHD), stroke (both ischemic and hemorrhagic), chronic obstructive pulmonary disease (COPD), lung cancer (LC), and Type II diabetes (T2DM) among adults (aged 25 and above), as well as acute lower respiratory tract infections (ALRI) among children (under 5 years old).

- **Line 264:** Add *"the"* before *"concentration"*.
  Thank you. We have added *"the"* before *"concentration"*. The revised sentence now reads: $RR_{c,a,d}$ is the Relative Risk, where c,a,d denotes the concentration of $PM_{2.5}$, population age, and disease, respectively; $BM_{a,d}$ is the baseline mortality rate per 100,000 population; and $P_a$ is the exposed population in a grid by age.

- **Line 264:** Add a comma before *"respectively"*.
  Thank you. We have added a comma before *"respectively"*. The revised sentence now reads: $RR_{c,a,d}$ is the Relative Risk, where c,a,d denotes the concentration of $PM_{2.5}$, population age, and disease, respectively; $BM_{a,d}$ is the baseline mortality rate per 100,000 population; and $P_a$ is the exposed population in a grid by age.

- **Line 265:** *"Age specific RRs (Relative Risks) for IHD and stroke, are obtained using MR-BRT."* → Remove the comma after *"stroke"*: *"Age-specific RRs for IHD and stroke are obtained using MR-BRT."*

Thank you. We have remove the comma after *"stroke"*: The revised sentence now reads:  Age specific RRs for IHD and stroke are obtained using MR-BRT.

- **Line 265:** *"For LC, T2-DM, and COPD uniform RRc,d were used across all age groups among adults."* → Add an article: *"For LC, T2DM, and COPD, a uniform RRc,d was used across all age groups among adults."*

Thank you. We have added an article. The revised sentence now reads: For LC, T2-DM, and COPD, uniform $RR_{c,d}$ were used across all age groups among adults.

- **Line 275:** *"For the concentration data generated using SSP emissions, we apply the respective SSP projections..."* → For consistency in verb tense: *"we applied the respective SSP projections..."*

We revised the sentence to: For the concentration data generated using SSP emissions, we applied the respective SSP projections for baseline mortality and population.

- **Line 294:** *"Notably, there are projected declines in BC emissions after mid- century under SSP245 and SSP245."* → There is a repetition of "SSP245." Based on Figure 2, it seems this should refer to "SSP585." Please verify and correct the second scenario accordingly.

Thank you very much for pointing that out. We've revised the sentence to: After mid-century, BC emissions in Africa are projected to decline under the SSP245 and SSP585 scenarios, driven by a reduction in residential fossil fuel use (Turnock et al., 2020).

- **Line 341:** *"Western Africa and Eastern"* → Suggested revision: *"Under the UNEP baseline scenario, BC, OC, and NOx emissions are projected to be largest in Western Africa and Eastern Africa, followed by Central Africa, Southern Africa, and Northern Africa, by 2050..."*
We revised the sentence to: Under the UNEP baseline scenario, BC, OC, NOx emissions are projected to be largest in Western Africa and Eastern Africa, followed by Central Africa, Southern Africa, and Northern Africa, by 2050, while $SO_2$ emissions are projected to be largest in Southern Africa, followed by Eastern Africa, Northern Africa, Western Africa, and Central Africa (**Table S3**).

- **Line 397:** *"The SSPs also demonstrates"* → Should be *"demonstrate"* (plural subject).

Thank you. We have removed this sentence entirely, as it was not directly relevant to the context of that section and its removal also addresses the first reviewer's suggestion to condense the text in that section.

- **Line 407:** *"Additionally, Under the ECL6 CLE scenario"* → The word *"Under"* should not be capitalized: *"Additionally, under the ECL6 CLE scenario..."*
Thank you. We've revised the sentence to: Additionally, under the ECL6 CLE scenario, by 2050, the industrial sector and the energy sector are projected to be the dominant contributors to the increase in $SO_2$ emissions in sub-Saharan Africa, except in Southern Africa, where $SO_2$ emissions are projected to decrease, with the industrial sector accounting for 56 % of this reduction.

- **Line 529:** *"even in scenarios projecting modest PM2.5 decreases (ECL6 SDS, SSP119, UNEP 2063, etc)"* → The use of *"etc."* is too vague for academic writing. If possible, specify the remaining scenarios or remove the phrase altogether.

Thank you for pointing that out: we have now listed the remaining scenarios: the revised sentence now reads: Due to large increases in population size in all African regions, PM$_{2.5}$-attributed mortality is projected to increase in all scenarios between the baseline and near future, even in scenarios projecting modest PM$_{2.5}$ decreases (ECL6 SDS, ECL6 MFR, SSP119, SSP585, and UNEP 2063) and strong decreases in baseline disease rates (depicted by prevalent negative blue bars for epidemiological transitions in **Figure S13**).

- **Line 531:** *"This further highlight that there are already high PM2.5 concentrations"* → Should be *"highlights"* to match the singular subject *"This."*

  Thank you. This sentence has been removed based on first reviewer comments

- **Line 545:** *"the RFari in 2050 relative to the baseline is negative in the UNEP scenarios, SSP119, and ECL6 SDS scenarios."* → To avoid repeating *"scenarios"*, consider rephrasing: *"the RFari in 2050 relative to the baseline is negative in the UNEP, SSP119, and ECL6 SDS scenarios."* Thank you for the suggestion. We agree with the interpretation but have clarified the sentence to reflect that the UNEP category includes the 3 UNEP scenarios. The revised sentence now reads: Averaged over the entire continent, the RFari in 2050 relative to the baseline is negative in SSP119, ECL6 SDS, and the UNEP scenarios, and positive in the remaining scenarios, reflecting a small cooling or warming effect depending on the scenario.

- **Line 554:** *"This shows the dominating influence of changes in scattering aerosols in the scenarios."* → Consider replacing *"dominating"* with *"dominant"*, which is more natural in this context.

  Thank you. We've revised the sentence to: This shows the dominant influence of changes in scattering aerosols in the scenarios.

- **Line 559:** *"Since emissions of changes in scattering aerosols dominate, the result is a positive radiative forcing."* → The phrase *"emissions of changes"* sounds awkward. A clearer option would be: *"Since changes in scattering aerosol emissions dominate, the result is a positive radiative forcing."*

  Thank you. We've revised the sentence to: *"Since changes in scattering aerosol emissions dominate, the result is a positive radiative forcing."*

- **Line 561:** Add *"the"* before *"relative importance of different aerosol emissions."*

  Thank you. We have added *"the"* before *"relative importance of different aerosol emissions." The revised sentence now reads:* The regional pattern of radiative forcing reflects not only the overall stringency of air pollution control in the scenarios but also subsequent changes in the relative importance of different aerosol emissions.

- **Line 566:** *"Hence, the negative radiative forcing is therefore nearly as strong in UNEP SLCP as in UNEP BASE."* → *"Hence"* and *"therefore"* are redundant. Consider removing one for clarity.

  Thank you. We've revised the sentence to: Hence, the negative radiative forcing is nearly as strong in UNEP SLCP as in UNEP BASE.

- **Line 570:** Typo: *"ECLE SDS"* should be corrected to *"ECL6 SDS."*

  **Thank you.** *ECLE SDS has been corrected "ECL6 SDS."*

- **Line 582:** *"lack of detailed assumptions"* → Consider rephrasing to *"lack of detailed assumptions in some cases"* to make the meaning clearer, as *"lack of assumptions"* can be ambiguous.

  *Thank you. We've revised the sentence to: The scenarios are designed to have large spread, reflecting uncertainties in how African nations will develop and lack of detailed assumptions in some cases.*

- **Line 585:** *"there is the need for improved data"* → For smoother phrasing, revise to: *"there is a need for improved data."*

  *Thank you. "the" has been changed to "a"*

- **Line 588:** *"where "real-world' emissions"* → The closing quote is incorrect. It should read: *"where 'real-world' emissions."*

  *Thank you. We've revised the sentence to: For instance, BC emissions show variability throughout the historical period, but $SO_2$ emissions exhibit more consistency between inventories, particularly in recent years, where 'real-world' emissions track the high projections.*

- **Line 594:** *"Bonjour et al. (2013) and Chowdhury et al. (2023), noted"* → Remove the comma after *"2023"*: *"Bonjour et al. (2013) and Chowdhury et al. (2023) noted..."*

  *Thank you. The comma has been removed. The revised sentence now reads: Additionally, Bonjour et al. (2013) and Chowdhury et al. (2023) noted that over 90 % of households in sub-Saharan Africa rely on solid fuels for cooking and domestic activities, further emphasizing the dominant role of the residential sector in BC emissions across all scenarios for the region.*

- **Line 685:** Replace *"is"* with *"are"* in reference to *"assumptions"*, which is a plural noun.

  *Thank you. We have replaced "is" with "are". The revised sentence now reads: Better and more spatially resolved assumptions about policy, technology, and economic development are needed for projections on the continent, especially for the upcoming IPCC AR7.*

- **Line 717:** *"by 2050 except UNEP BASE and UNEP SLCP"* → For clarity and smoother flow: *"by 2050, except in the UNEP BASE and UNEP SLCP scenarios."*

  *Thank you. We've revised the sentence to: We find a net positive aerosol-induced forcing across sub-Saharan Africa in all scenarios, by 2050, except UNEP BASE and UNEP SLCP scenarios, with values ranging from 0.03 W m$^{-2}$ in SSP119 to 0.27 W m$^{-2}$ in SSP585, driven mainly by changes in scattering aerosols.*

- **Line 723:** *"across Sub-Saharan Africa"* → Within a sentence, *"sub-Saharan Africa"* should be in lowercase.

  *Thank you. We have changed "S" to "s" in Sub-Saharan Africa*

- **Line 734:** *"Accurate activity data and harmonization efforts are especially essential for upcoming assessment efforts"* → Consider rewording to avoid the repetition of *"efforts"*, e.g., *"Accurate activity data and harmonization are especially essential for upcoming assessments."*

  *Thank you we have revised the sentence to: Accurate activity data and harmonization are especially essential for upcoming assessments aimed at informing policy makers, such as the various elements of the 7th Assessment Report of the Intergovernmental*

Panel on Climate Change, which will rely on these baselines to effectively consider the implications of future climate and air quality policies.

---

## Referee Report (RR1)

The authors have substantially changed the paper to address the comments raised in review, accounting for each comment in turn. In particular, Section 3.1 is now much more easy to follow, Section 3.7 has the detail it deserves, and the Figures are much better – especially 3 and 4. There are some small clarifications that need to be made just to iron things out, and there is one conceptual issue regarding the definition of the "demographic transition" in terms of aging versus overall population growth. In light of the substantial alterations to address comments, and the very minor changes that need making, I recommend this revision be accepted subject to technical corrections, which do not need further review.

Figure 1: this can be cropped at -40 latitude to remove the white space. Also needs a full stop in the caption.

Section 2.2: good to have the resolutions, but would also just highlight around L210 that they all have the same 0.5x0.5, to aid the reader.

L262 needs to be ", and ammonium" in the list.

L280: need citation for "Natural Earth dataset"

L332: "ranging from 2.4 µg m-3" to what? (5.9, uniformly, I think? Need to clarify both the range and the distribution ie uniform or other). Also, how many samples? It's fine to refer to the other paper for more specific details but the key info should be given here, and I think that should include the number of samples.

L351: at the end you have "a grid by age." which highlights the dependence of the population, but you don't do the same for the mortality or RR. Maybe for clarity add a new sentence noting which properties each variable depends on.

L362: "about 25 km" – why is this only "about"? Is the resolution in degrees and this is approx? It's not too important but worth clarifying.

L407: "For instance, [in] Central Africa"

Figs 3+4: these are much better, but 1) cape verde encroaches on the text – maybe just make the boxes non-transparent 2) I feel the boxes could possibly be rearranged to the sides, to make the figure wider and allow for it to be bigger overall therefore – but if you have played around with it and this is the best configuration then that's fine.

Fig 5: much better; caption needs a full stop.

L605: this is substantially improved and really interesting! However, there's some confusion around the "demographic transition" and aging. The "demographic transition" is comprised of aging and general population increase, right? But you attribute the demographic transition bar in S13 purely to aging in the text. Really, to explore this effect, you'd have to repeat the analysis with the population increased but the age distribution constant, versus the results where both change, i.e. decompose the demographic transition into aging and population growth. I'm not suggesting you need to do this unless you're interested in exploring this aspect, but you at least need to resolve this ambiguity by not referring purely to aging as the cause of these deaths. I also think S13 is a key figure and deserves to be in the main text really, unless there's a limit on figures here. But up to you.

---

## Author Response (AR2)

**Note from the Production office**:
Coloured or marked text in *.pdf manuscript file is not allowed (page 9, Eq 1 ). Please provide a
clean version of *pdf manuscript file (without marked changes, but with black text) for next
revision.

Thank you for the comment. We have now provided a clean version of the manuscript PDF with
all text, including Equation 1 on page 9, in black and without any marked changes.

**Reviewer's comments:**

 The authors have substantially changed the paper to address the comments raised in review,
accounting for each comment in turn. In particular, Section 3.1 is now much more easy to follow,
Section 3.7 has the detail it deserves, and the Figures are much better – especially 3 and 4.
There are some small clarifications that need to be made just to iron things out, and there is one
conceptual issue regarding the definition of the "demographic transition" in terms of aging
versus overall population growth. In light of the substantial alterations to address comments,
and the very minor changes that need making, I recommend this revision be accepted subject to
technical corrections, which do not need further review.

We sincerely thank the reviewer for the thoughtful and constructive feedback throughout the
review process. We are very pleased that the revised manuscript is now clearer and that the
improvements to Section 3.1, Section 3.7, and Figures 3 and 4 were well received.

We have carefully addressed the remaining minor clarifications and resolved the conceptual
issue regarding the definition of the "demographic transition" to ensure accuracy and
consistency throughout the manuscript.

We appreciate your recommendation for acceptance with technical corrections and are grateful
for your support in bringing this work to publication.

Figure 1: this can be cropped at -40 latitude to remove the white space. Also needs a full stop in
the caption.

Thank you for the helpful suggestion. We have now cropped the figure at –40° latitude to
remove the excess white space, and we have also added a full stop at the end of the caption as
recommended.

Section 2.2: good to have the resolutions, but would also just highlight around L210 that they all
have the same 0.5x0.5, to aid the reader.

Thank you for your positive feedback on the scenario selection description. We agree that
highlighting the common spatial resolution will help readers, and we have added a sentence
around L210 clarifying that all model datasets are at a consistent 0.5° × 0.5° resolution.

We added the following to the manuscript:

All emissions datasets used share a common horizontal resolution of 0.5° latitude by 0.5°
longitude, facilitating direct comparison and integration across scenarios.

L262 needs to be ", and ammonium" in the list.

Thank you for pointing this out.

We have added ", and ammonium" to L262 in the manuscript, The sentence now reads: $PM_{2.5}$ was calculated as the sum of individual fine-mode aerosol species, namely BC, primary and secondary organic aerosol (POA, SOA), sulfate ($SO_4$), dust, sea salt, nitrate ($NO_3$), and ammonium ($NH_4$).

L280: need citation for "Natural Earth dataset"

Thank you for your suggestion. We have added a citation, and the revised sentence now reads: "Grid cells were assigned to countries based on whether their center point lies within national boundaries, using a shapefile of administrative boundaries from the Natural Earth dataset (Natural Earth, 2025)."

L332: "ranging from 2.4 µg m-3" to what? (5.9, uniformly, I think? Need to clarify both the range and the distribution ie uniform or other). Also, how many samples? It's fine to refer to the other paper for more specific details but the key info should be given here, and I think that should include the number of samples.

Thank you for your comment. We have clarified the text to indicate that the counterfactual concentration (TMREL) was held constant at 2.4 µg m$^{-3}$ in this study, rather than distributed across a range. We have also ensured that this is clearly stated in the manuscript (L332).

The revised sentence reads: We assumed a theoretical minimum risk exposure level (TMREL) of 2.4 µg m$^{-3}$, held constant across all locations and scenarios (Murray et al., 2020).

L351: at the end you have "a grid by age." which highlights the dependence of the population, but you don't do the same for the mortality or RR. Maybe for clarity add a new sentence noting which properties each variable depends on.

Thank you for your suggestion. We have added the following after "a grid by age.": "Each variable is a function of specific dimensions: RR varies with concentration of $PM_{2.5}$, age group, and disease; BM varies with age group and disease; and population varies by age group and location"

L362: "about 25 km" – why is this only "about"? Is the resolution in degrees and this is approx? It's not too important but worth clarifying.

Thank you for your suggestion. The resolution is 25 km × 25 km, not an approximation. We have therefore removed the word "about" to improve clarity.

The revised sentence now reads: Age distributions at 5-year intervals (adults > 25 years), and <5 years for children were obtained from the SSP database (Riahi et al., 2017) which are then merged with the gridded population data at 25 km by 25 km horizontal resolution under the respective SSP scenarios to obtain the age-specific population ($P_a$) at each 25 km by 25 km grid.

L407: "For instance, [in] Central Africa"

Thank you for pointing this out. We have added "in" to the sentence, so it now reads "For instance, in Central Africa..." in the revised manuscript.

Figs 3+4: these are much better, but 1) cape verde encroaches on the text – maybe just make the boxes non-transparent 2) I feel the boxes could possibly be rearranged to the sides, to make the figure wider and allow for it to be bigger overall therefore – but if you have played around with it and this is the best configuration then that's fine.

Thank you very much for your helpful suggestion. We experimented with making the boxes non-transparent; however, this caused them to completely obscure Cape Verde. After careful consideration, we believe that the current configuration represents the best balance between readability and geographic clarity. Thank you for understanding.

Fig 5: much better; caption needs a full stop.

Thank you for your positive feedback on Figure 5. We have added the full stop at the end of the caption as requested.

L605: this is substantially improved and really interesting! However, there's some confusion around the "demographic transition" and aging. The "demographic transition" is comprised of aging and general population increase, right? But you attribute the demographic transition bar in S13 purely to aging in the text. Really, to explore this effect, you'd have to repeat the analysis with the population increased but the age distribution constant, versus the results where both change, i.e. decompose the demographic transition into aging and population growth. I'm not suggesting you need to do this unless you're interested in exploring this aspect, but you at least need to resolve this ambiguity by not referring purely to aging as the cause of these deaths. I also think S13 is a key figure and deserves to be in the main text really, unless there's a limit on figures here. But up to you

Thank you for this insightful comment. We agree that the demographic transition encompasses both aging and overall population growth.  In the revised text, we have refrained from attributing the demographic transition effects solely to aging and have instead highlighted the combined influence of both aging and population growth on excess deaths. We acknowledge that fully decomposing these effects by holding one factor constant while varying the other would be valuable, though it is beyond the scope of the current study. We revised sentences in L605 paragraph as follows:

"Aging and population growth are both key aspects of the demographic transition, each contributing to increased excess deaths. Aging increases excess deaths from age-related diseases, even in scenarios with modest population growth. In parallel, overall population growth amplifies excess deaths, particularly in regions with high fertility rates and younger populations, such as Western Africa and Eastern Africa. In Northern Africa, while population growth is projected to be slower, the aging population resulting from demographic shifts is projected to lead to a substantial number of excess deaths, reaching 300,000 under SSP119 and SSP585, and exceeding 200,000 in other scenarios (**Figure 8**). Southern Africa is projected to experience the lowest number of excess deaths attributable to aging and population growth,

except under SSP119 and SSP585, where demographic transition-related excess deaths increase to about 20,000."

Thank you for the helpful suggestion regarding Figure S13. We agree that it presents important results. In response, we have now included this figure in the main manuscript as **Figure 8**,and revised the figure numbers throughout the paper accordingly to reflect this change. Thank you again for your helpful feedback.